



# Mineral surface area in deep weathering profiles reveals the interrelationship of iron oxidation and silicate weathering

Beth A. Fisher[1], Kyungsoo Yoo[2], Anthony K. Aufdenkampe[3], Edward A. Nater[2], Joshua M. Feinberg[4], Jonathan E. Nyquist[5]

[1]Biochemistry, Chemistry, and Geology, Minnesota State University, Mankato, Mankato, 56001, USA
[2]Soil, Water and Climate, University of Minnesota, St. Paul, 55108, USA
[3]LimnoTech, Oakdale, 55128
[4]Earth & Environmental Sciences, University of Minnesota, Minneapolis, 55455, USA
[5]Earth & Environmental Sciences, Temple University, Philadelphia, 19122, USA

*Correspondence to*: Beth A. Fisher (beth.fisher@mnsu.edu)

**Abstract.** Mineral specific surface area (SSA) is generated as primary minerals weather and restructure into secondary phyllosilicate, oxide, and oxyhydroxide minerals. SSA is a measurable property that captures cumulative effects of many physical and chemical weathering processes in a single measurement and has meaningful implications to many soil processes, including water holding capacity and nutrient availability. Here we report our measurements of SSA and

mineralogy of two 21 meter deep SSA profiles at two landscape positions, in which the emergence of a very small mass percent (0.8-2.7%) of secondary oxide generated 36-81% of the total SSA at both landscape positions. The SSA transition occurred at 3 meters and did not coincide with the morphological boundaries of soil to weathered rock or with the water table. The 3 meter boundaries coincide with the depth extent of secondary iron minerals and secondary phyllosilicates. Although elemental depletions in both profiles extend to 7 and 10 meters, secondary minerals were not detected below 3

meters. The 3 meter depth marks the emergence of secondary oxide minerals, and this boundary appears to be the depth extent of oxidation weathering reactions. Our results suggest that oxidation weathering reactions may be the primary limitation in the coevolution of both secondary silicate and secondary oxide minerals.

## 1 Introduction

Many biogeochemical and environmental reactions in terrestrial and aquatic ecosystems occur on the surfaces of minerals.

Specific surface area (SSA) of minerals is relevant to the capacity of soil and sediment to store moisture, stabilize organic carbon, and regulate mineral dissolution. Research communities interested in the mechanisms of soil organic carbon stabilization recognize that interactions between organic matter and minerals may be a critical protective mechanism for organic matter in soils (Aufdenkampe et al., 2011; Berhe et al., 2007; Fisher et al., 2017a; Kleber et al., 2007; Schmidt et al., 2011; Wang et al., 2018; Yoo et al., 2011). The marine sediment community has long recognized the utility of SSA as a

proxy to understand the potential for mineral surfaces to stabilize associated organic matter (Hedges et al., 1999; Keil and Mayer, 2014; Mayer, 1994). Advancing our understanding of processes that influence the production and distribution of



mineral surfaces has implications for a broad range of questions related to critical zone evolution, structure, and function. Here our overarching goal was to measure the development of mineral SSA during evolution of the critical zone of the earth and identify the processes and sources of SSA development.


The critical zone extends from the top of the vegetation canopy to the deepest limits of actively cycling groundwater (Brantley et al., 2007). This upper layer of the earth has been termed the "critical zone" to facilitate interdisciplinary research questions and methods on interdependent hydrological, geochemical, geomorphic, biological and other processes. A consequence of this interdisciplinary focus is that the many functional layers of the critical zone are often defined using

different terms by different disciplines. Acknowledging this difficulty, we provide the following definitions for the three-layered system we use to describe the measured and observed features and landforms in the present study.

We define bedrock as rock that has not been subject to alteration by physical or chemical weathering. We use the term weathered rock to describe the part of bedrock that has been chemically weathered but has not been physically mixed or

mobilized, as evidenced by the retention of rock structure.

We consider weathered rock to include both isovolumetric weathered rock and weathered rock with volumetric alteration that has not been mixed or mobilized. These components of weathered rock are commonly referred to as saprock, saprolite and/or immobile regolith, with definitions of these terms often differing slightly by author (Graham et al., 2010; Lebedeva et

al., 2010). We also consider weathered rock to include the C horizon, which is referenced by pedologists to describe the zone below which pedogenic alterations are no longer evident by field identification (NRCS, 1993).

We use the term soil to describe the physically mobile layer above weathered rock, in which physical and chemical weathering processes are most active, and which develops genetic horizons (NRCS, 1993). We consider soil to be

synonymous with what geomorphologists recognize as the mobile regolith, which describes material that has been physically mixed or displaced and no longer retains rock structure (Anderson et al., 2013).

Although the transitions between these three layers can be abrupt, they most often occur as gradual transitions, and for brevity of discussion we will term the transitions as boundaries. The boundaries between soil, weathered rock, and fresh

bedrock are shaped by complex interactions of geomorphic, geochemical, biological, and hydrological processes. These boundaries have been most actively studied at the hillslope scale (e.g., Rempe and Dietrich, 2014; Hasenmueller et al., 2015), where the hillslope is selected because it is the fundamental unit of landscapes (Carson and Kirkby, 1972). At the soil-weathered rock boundary (

Figure 1), consolidated rock is physically and chemically broken down to unconsolidated soil that may be subject to gravity-

driven colluvial transport (Carson and Kirkby, 1972; Dietrich et al., 1995; Heimsath et al., 2005). The conversion from



weathered rock to soil is a quantifiable process and has been termed soil production (Heimsath et al., 1997) where the chemical weathering rate within the soil is directly proportional to soil production rates (Gilbert, 1909; Raymo and Ruddiman, 1992; Riebe et al., 2004; West et al., 2005).

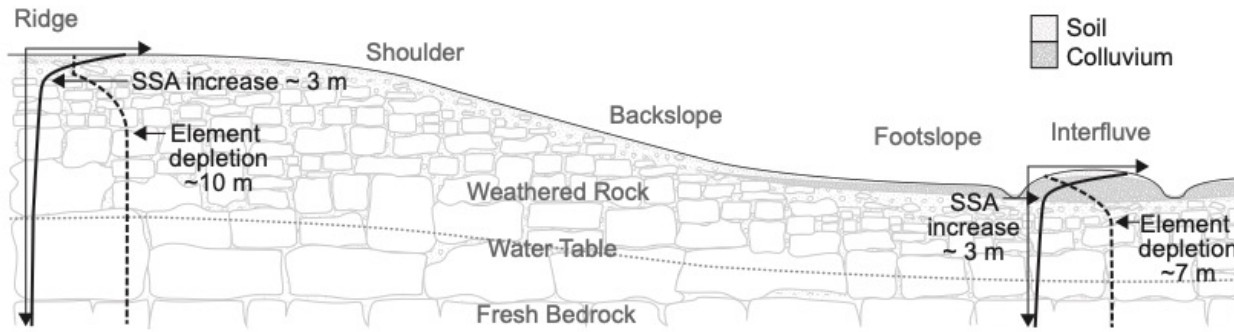


**Figure 1. Schematic figure of study results superimposed on a landscape cross section illustrating the study site hillslope positions and zero order interfluve. The interfluve, distinct from the footslope, is a convergent area between two ridges that was once a collection point for colluvium and has since been eroded to a convex hill bounded by gullies. The conceptual model illustrates a three layer model of soil, weathered rock, and fresh bedrock. We show our SSA profiles alongside results of elemental depletion of**
**calcium from a parallel study** (Fisher et al., 2017b)**. We found abrupt transitions in surface area at ~ 3 m depth at both landscape positions. Elemental depletion extended ~ 10 m at the ridge and ~ 7 m at the interfluve.**

The weathered rock to bedrock transition can be identified using several process based approaches. The deepest reach of
rock that has been influenced by weathering, or the weathering front, is understood to be a downward propagation of chemical weathering processes (Brantley et al., 2013a). Weathering fronts occur due to the equilibrium status of pore waters (Maher, 2010), the acidity of pore waters (Brantley et al., 2013b; Hasenmueller et al., 2015), and gas weathering interactions (Kim et al., 2017; Stinchcomb et al., 2018). Synthesis models hypothesize that the depth of the groundwater table sets a limit and lower boundary for weathering profiles (Rempe and Dietrich, 2014). Weathering may terminate at the groundwater table
due to slow moving groundwater reaching equilibrium with bedrock (Rempe and Dietrich, 2014) or the penetration of acidity into the formation, which may be exhausted at depths that coincide with the groundwater table (Brantley et al., 2013b). Temporal fluctuation of the groundwater table, in combination with redox cycling may control the release of iron from primary minerals and subsequent precipitation of secondary oxides (Haberer et al., 2015). Geochemical and geophysical methods have been used to identify or image these deep critical zone transitions (e.g. Holbrook et al., 2014; Parsekian et al.,
2015; Perron et al., 2015). In this study we explore the boundaries of weathering profiles with SSA.



## 2 Hypothesis

Studies and models of chemical weathering suggest that morphologic boundaries should coincide with observable process based transitions in weathering and pedogenesis (Bazilevskaya et al., 2013; Brantley et al., 2013b; Brantley and Lebedeva, 2021; Brantley and Olsen, 2013; Gu et al., 2020; Hasenmueller et al., 2015; Maher, 2010; Parsekian et al., 2015; Pedrazas et

al., 2021; Rempe and Dietrich, 2014; Riebe et al., 2017). Thus, we expected to observe changes in mineral SSA at morphologic boundaries, recognizing that SSA of bulk soil or rock is highly sensitive to small changes in the presence and abundance of secondary phyllosilicate minerals and iron oxides. We hypothesized that SSA would develop from the acid driven weathering that generates secondary phyllosilicate minerals and that SSA would therefore begin increasing at the same weathering fronts at around 7-10 m identified using element depletion by Fisher et al. (2017a), as shown in Figure 1.

Instead we observed negligible increases in SSA from bedrock at around 20 m to the weathered rock at 3 m followed by a sharp increase in SSA from 3 m to the ground surface. We did not observe significant changes in SSA at the typical boundaries that that receive attention in weathering models: soil, weathered rock, bedrock, water table, etc.

To test the hypothesis that SSA production would increase at the onset of elemental depletion, we characterized the SSA,

mineralogy, and geochemistry of material from two 21 meter cores from a 100% forested watershed underlain by schist. In addition to drilled cores at ridge top and a zero order interfluve, which occurs ten meters below the ridge, we collected seismic data between drilled cores. The interfluve is distinct from the footslope and is a local convergent area that was once filled with colluvium and has since been eroded and incised to form a convex hill bounded by fluvial gullies (

Figure 1 and Figure 2). Efforts to measure chemical weathering transitions as a function of hillslope topography using

geophysical and geochemical methods abound, but we know of no previous efforts to measure the vertical and lateral distribution of SSA in a landscape, through entire weathering profiles and below the water table at both eroding and depositional localities.



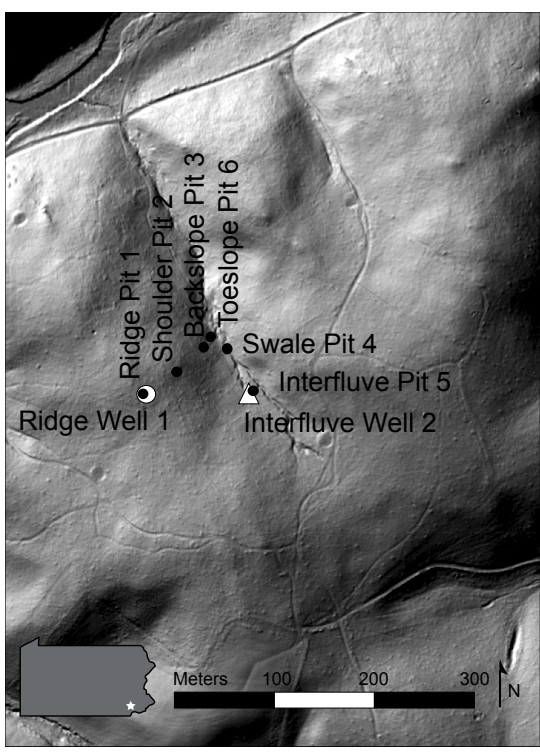

**Figure 2. Ground returns Lidar hillshade image provides a site map with topographic information of the Spring Brook watershed in the Laurels Preserve in southeastern, Pennsylvania. In addition to Ridge Well 1 and Interfluve Well 2, we sampled a transect of soil pits to characterize the influence of landscape position on SSA and other biogeochemical properties** (Fisher et al., 2017a).

## 3 Study Site

The study site is the Spring Brook watershed (Figure 2) located in southeastern Pennsylvania and within the Christina River Basin Critical Zone Observatory (CRB-CZO). The bedrock in the site is the Laurels Schist, a foliated, silvery, gray-green, quartz, plagioclase, muscovite, chlorite, garnet schist with minor biotite (mostly retrograded to chlorite) and accessory magnetite, epidote, tourmaline, apatite, and zircon (Blackmer, 2004). The foliation in the Laurels Schist weathers to high-density, platy segments of rock that remain virtually unweathered internally. Weathering occurs primarily along fractures and foliations between the plates. The soils in Spring Brook are mapped as Manor Series soils (NRCS, 2012). These Typic Dystrudepts are highly micaceous with weak structures and coarse sandy clay loam textures. The abundant coarse fragments in all of the soils in Spring Brook are composed of angular platy rock fragments of schist called channers. Detailed soil characteristics are outlined in Fisher et al., 2017a.

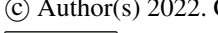



Spring Brook is a first order watershed that covers 9.6 ha, with a 250 m long spring fed perennial stream. Up gradient from
the stream is a 150 m long, 1 m deep historic gully that is no longer actively eroding and is a depositional swale that has been
stable long enough to develop an A horizon with significant organic matter accumulation throughout the extent of the swale.

The Spring Brook watershed is a second growth mixed chestnut, oak, and hickory forest of approximately 120-150 years in
age, and was likely logged for 1-2 cutting and planting cycles of 25-30 year old hardwood trees to provide charcoal for local
smelting operations beginning around 1825 (Lesley, 1859; Shields and Benson, 2011). Remnants of charcoal production are
evident in the landscape and appear as small (2-3 m diameter) level, circular features in lidar imagery (Figure 2). The
property was purchased for hobby farming in the early 1900's (did not involve tillage), was used for cattle grazing beginning
in 1946. In 1967 the Brandywine Conservancy dedicated the property as the Laurels Preserve and ended grazing activity
(Shields and Benson, 2011).

The climate in Spring Brook is humid continental with mean annual precipitation (MAP) of 1246 mm and a mean annual
temperature (MAT) of 10.85°C. The full annual temperature range was -6 to 29°C (1961-1990, Coatesville, PA,
www.usclimatedata.com). Paleoclimate records based on pollen type between 18,000-12,000 ybp estimate annual lows of -
12°C and highs of 16°C, which represents a periglacial time period prior to Laurentide glacial retreat. By 9,000 ybp the
temperatures were -4 to 22°C, with a gradual shift to present day temperature ranges (Prentice et al., 1991).

## 4 Methods

### 4.1 Rotosonic drilling

To capture the subsurface topography of weathered materials in Spring Brook, we selected the ridge top (Well 1, 143.886
MASL; 39.9195025°, -075.7891562°; IGSN:IESW10006) and interfluve (Well 2, 134.164 MASL; 39.9194885°,
-075.7879179°; IGSN:IESW10007) for drilling. Two 21 m deep boreholes were drilled into the Laurels Schist formation in
August 2012. Drilling of additional boreholes was precluded by logistical and financial challenges in the remote, densely
forested, steep terrain. Samples were acquired using a Geoprobe Rotosonic (model 8140LC) mid-sized track-mounted
drilling rig. The drilling method employed was the "4x6" method, which involves two hollow bits that yield a 4 inch (10.16
cm) sample diameter and a 6 inch (15.24 cm) borehole diameter. Sample intervals up to 10 m in Well 1 and to 5 m in Well 2
were drilled with air (i.e. no fluids) to maximize recovery and nearly eliminate contamination. Deeper intervals alternated
between no drilling fluid and EZ-MUD® Polymer Emulsion (by Baroid IDP) to enable drilling to proceed in a timely
manner. The drill segments were 1.52 m and were inserted into a plastic sleeve immediately after recovery. While the cores
were partially pulverized by the Rotosonic drilling action and segmented due to the foliated nature of the schist, the
recovered volumes were 68-100% for core segments drilled without fluid. Where EZ-MUD fluid was used, the recoveries



ranged from 17-83%, and contained large rock pieces with no pulverized material. Missing sample intervals in the data set occur where segments were drilled with fluid. Drilling progress was hindered in Well 2 when the drill bit broke after approximately 12 meters of recovery. In the laboratory the cores were removed from their plastic sleeves, photographed (supplementary file), and divided into intervals of approximately 10 cm. Samples were oven dried at 60°C, sieved to 2 mm,
and weighed. The drilled wells were maintained with flexible liners (FLUTe™ Blank liners) that can be removed for measurements and future installations.

## 4.2 Soil sampling

In addition to drilling, soil pits were hand excavated to ~1 m deep within a few meters of (Pit 1, ridge: 143.932 MASL, 39.9194572°, -075.7892333°, IGSN:IESW10001; Pit 2, shoulder: 139.949 MASL, 39.919783°, -75.7888499°,
IGSN:IESW10002; Pit 3, planar backslope: 129.989 MASL, 39.919926°, -75.7885518°, IGSN:IESW10003; Pit 6, toeslope: 123.096 MASL, 39.9200122°, -75.7883606°, IGSN:IESW10005; Pit 4, swale: 39.919863°, -75.788162°, IGSN:IESW1001C; Pit 5, interfluve: 133.108 MASL 39.9195557°, -075.7878866°, IGSN:IESW10004). After detailed soil description (NRCS, 2012), soil sample collection was guided by morphological horizons. Soil materials were sampled across the entire width of the upslope side of the soil pit to integrate heterogeneities. Soil samples were homogenized in the lab,
oven dried, and sieved to 2 mm, and weighed. We did not collect forest litter.

## 4.3 Specific surface area (SSA)

Samples from the fine fraction (<2 mm), which was mostly generated during grinding associated with drilling, were retained for measurement of SSA. Oven dried (60°C) samples were degassed at 150°C for a minimum of four hours in $N_2$ saturation (Mayer and Xing, 2001), weighed, and analyzed using $N_2$ adsorption on a Micromeritics TriStar II 3020. Specific surface
area (mineral surface area per unit mass of soil) was calculated with an 11 point isotherm using the BET multipoint isotherm method (Brunauer et al., 1938). For each sample, we measured SSA on three sample treatments to collect information about organomineral associations and Fe oxides. First, untreated samples that may contain organic matter were measured (Dataset S1), then the samples were oxidized at 350°C for 12 hours to remove organic matter (Keil et al., 1997), which is the most widely used method to prepare mineral surfaces for $N_2$ adsorption (total SSA). It is well understood that some mineral
transformations can occur under these conditions, but tests by multiple researchers show that these treatments minimally alter total SSA (Keil et al., 1997). For example, our tests of the effects of heating on three soil types and synthesized iron minerals revealed that SSA increased with increased heating temperatures from 150 to 350°C (unpublished lab results). Regardless, all samples for our study were subjected to the same pretreatments, so internal comparison is not a concern. Finally, Fe and Al oxides were removed by citrate-dithionate method (Burt, 2004) to measure the SSA of primary and
secondary silicate minerals ($SSA_{si}$) and calculate the contribution of extractable oxides to SSA as the fraction of $Fe_d$ SSA, where $Fe_d$ SSA$=1-(SSA_{si}/SSA)$. In sample intervals where replicate measurements occurred, the mean is reported.





SSA of naturally weathered minerals (e.g. sheet silicates or needle like oxides) substantially exceeds that of mechanically

ground minerals (Brantley and Mellott, 2000; Knauss and Thomas J, 1989; White et al., 1996). To confirm that partial

pulverization from Rotosonic drilling did not interfere with our estimate of naturally derived SSA, we analyzed SSA on

samples that were pulverized in a tungsten carbide ring mill to a particle size that is finer than that produced by Rotosonic

drilling. Rocks from Ridge Well 1 at 13 m and 16 m were pulverized to $D_{50} < 20$ μm (by laser particle size analysis) and

yielded SSA of 6.27 $m^2$ $g^{-1}$ and 6.37 $m^2$ $g^{-1}$ respectively. If the same particles were perfect cubes rather than mineral-shaped,

those samples would yield SSA of only 0.15 $m^2$ $g^{-1}$ (calculated with density 2600 kg $m^{-3}$). Rotosonic samples from the finest

samples from Ridge Well 1 at 13 m and 16 m were sand-sized and ranged in SSA from 3-5 $m^2$ $g^{-1}$. These results validate that

natural mineral structure dominates the SSA measured on drill pulverized samples.

**4.4 Inventory of total mineral surface area**

The total quantity of mineral surface area within a volume of a soil profile has not been previously measured to our

knowledge. We determine the cumulative inventory of total mineral surface area per given ground surface area (surface area

inventory, SAI, in unit dimensions of $L^2L^{-2}$) from the ground surface to the lower depth (z) limit of H as:

$$SAI = \int_{z=0}^{z=H}[SSA_z\rho_z(1-f_z)]dz \tag{1}$$

Where *SSA* is the total mineral specific surface area ($L^2M^{-1}$) in the depth *z*, $\rho$ is the bulk density of weathered material ($ML^{-3}$), and *f* is the coarse fraction (>2 mm, $MM^{-1}$). Because the depth to unweathered bedrock varies, we designed SAI to be

calculated to a depth of choice (i.e. z=H with units of L).


We calculate SAI beginning at the ground surface and increasing within lower depth limit because the surface is a natural

boundary and because we cannot assume the terminal depth of significant weathering and SSA generation. Furthermore,

calculating and visualizing inventory in this direction follows conventions used for soil organic carbon and cosmogenic

radionuclide inventories (e.g. Jobbágy and Jackson, 2000). The noteworthy points in the visualization are the slope and the

changes in slope, which enable us to identify process-driven transitions.

**4.5 Bulk density**

Bulk density measurements for Well 1 were directly measured from the ground surface to the depth of 3.41 m from a nearly

continuous set of 2.4 cm diameter cores that were collected from an excavated soil pit a few meters from the well. Bulk

density for Well 2 was measured using the same method for the first 1.80 meters. Soil cores were segmented in the lab at

intervals no greater than 10 cm, air dried, oven dried for 48 hours at 60°C and weighed. The USDA standard drying

temperature is 110°C (Burt and Staff, 2014), but we chose 60°C to minimize the degradation caused by heat on

phyllosilicates and metastable oxide minerals. The stability of these minerals was a concern for XRD analysis and other





biogeochemical measurements in the soil profile. Subsamples for SSA received additional treatment, as described in Section 3.3.


We were not able to directly measure bulk density from depths of 5 m to 21 m at Well 1 and from 4 m to 21 m at Well 2. For these intervals we estimated bulk density based on average rock fraction and density of rock chips. The specific gravity of rock chips was measured by water displacement: dry rock chips were weighed in air and the volume of the rock was determined by the mass of water displaced by the rock chips, with 2-5 replicates for each interval. Because rock chips had

low permeability, we did not coat them with plastic or wax, as proposed by Jin et al. (2010). Rock chip density values in each well did not vary systematically with depth. Average rock chip density for Well 1 (n=63) was 2700±100 kg m$^{-3}$ and for Well 2 (n=108) the average was 2770±110 kg m$^{-3}$. Where we have rock chip density measurements and bulk density cores in the top 3 meters, the bulk density was 53-69% lower than rock chip density. We know that the influence of weathering decreases with increasing depth, but we do not observe an increase in rock chip density with depth, so we set the bulk

density estimate as a constant. Thus, for deep bulk density estimates we used 68% of the rock chip density for each interval to capture the high end of the bulk density range observed from 0-3 m, which yields conservative underestimates in inventory calculations. The bulk density values below 5 m averaged 1830±70 kg m$^{-3}$ (n=25) in Well 1 and 1880±90 kg m$^{-3}$ (n=16) in Well 2.

**4.6 Characterization of bulk mineralogy**

To examine the links between mineral surface area and mineralogy, we characterized mineralogy of bulk samples using x-ray diffraction (XRD) on a Siemens D-500 Diffractometer with 2.2 kW sealed cobalt source at the University of Minnesota Characterization Facility. Samples were pulverized on a tungsten carbide ring mill to 6 μm (D$_{90}$) and combined with a 10% zincite internal standard (Eberl, 2003). The mixture was combined using an agate mortar and pestle wetted with ethanol and was packed into a holder and scanned from 5 (machine minimum) to 75° 2θ using Co Kα radiation with 0.02° steps and a

dwell time of 2 seconds per step on a continuously rotating sample stage.

XRD spectra were converted to Cu Kα wavelengths for easy comparison with earlier studies using JADE 7.0 software (Materials Data, Inc.) and exported as ASCII text for analysis. Peak intensities were manually normalized in spreadsheet form setting the zincite 2.476Å peak as 10%. Major minerals were identified and quantified using non-interfering peaks and

normalized by expected peak intensities: Quartz 1.8179Å (0.14 intensity), Plagioclase group 3.17-3.21Å (1), Muscovite 9.95Å (0.95), and Clinochlore (Chlorite group) 4.77Å (0.7).



### 4.7 Clay mineralogy

Given the expected disproportionate contributions of clay sized minerals to mineral SSA, we separately characterized their mineralogical compositions. The clay size fraction was isolated by gravity sedimentation using a suction apparatus (Jackson
and Barak, 2005). Isolated clay sized fractions were pretreated in multiple steps to remove carbonates (sodium acetate-acetic acid solution), organic matter (bleach method), and extractable iron oxides (citrate-dithionite) (Burt, 2004; Jackson and Barak, 2005). Because our samples were low in carbonate content and not extremely high in organic matter, we were not concerned that cementation would hinder clay isolation prior to pretreatment. We expect that if some clays were lost during fractionation, the ratios of different clays are likely to be maintained within the detection sensitivity of XRD.


Clays were divided and saturated with potassium or magnesium and were mounted by oriented suspension on glass coverslips. Saturated samples were scanned from 5 to 18° (K-saturated) and 5 to 36° (Mg-saturated) 2θ using Co Kα radiation. Potassium saturated samples were heated to 500°C for 1 hour and scanned again from 5 to 18° 2θ. Magnesium saturated samples were further saturated with ethylene glycol and scanned from 5 to 18° 2θ. Clay XRD methods may have
been impacted by pulverization by sonic drilling action, which broke primary minerals into pieces that are small enough to partition with clays in our gravity settling separation, and the pulverized particles may inhibit the secondary phyllosilicates from orienting parallel to the glass slides.

We have chosen to not quantify clay minerals in this study for the following reasons. Repeated chemical rinses and vacuum
separation of clay particles from the bulk sample may bias quantification of secondary phyllosilicate minerals. Additional bias may be introduced into the quantification of samples from greater depths collected from Rotosonic drill cores because pulverization by sonic drilling action broke primary minerals into pieces small enough to separate with clays, inhibiting the secondary phyllosilicates from orienting parallel to the glass slides.

### 4.8 Iron mineralogy

On a small subset of specimens from Well 1 (n=12), we used rock magnetic measurements to characterize the iron mineral type, quantity, and size (Dunlop and Ozdemir, 2007). Samples were pressed into 6 mm diameter pellets under pressure in a mixture of rock powder with SpectroBlend powder (81.0% C, 2.9% O, 13.5% H, 2.6% N). Room temperature and low temperature saturation isothermal remanence (RTSIRM, LTSIRM) was measured on five of the specimens using a Magnetic Properties Measurement System (MPMS, Quantum Designs). A room temperature Vibrating Sample Magnetometer (VSM,
Princeton Measurements) was used to obtain hysteresis loops for all samples. We applied a 1.25 T maximum magnetic field and measured bulk hysteresis parameters, saturation magnetization (Ms), saturation remanence (Mr), coercivity (Bc), and coercivity of remanence (Bcr). To isolate the magnetization of hematite and goethite, we sequentially removed the



magnetization associated with magnetite using a 200 mT alternating field demagnetization step and then removed the magnetization associated with goethite using a 100°C thermal demagnetization step.

## 4.9 Extraction chemistry

We used the citrate-bicarbonate-dithionite (CBD) method to remove free iron oxides from samples (Burt, 2004; Holmgren, 1967; Jackson and Barak, 2005). In each instance we applied this method on samples where organic matter was previously removed. In the CBD solution, iron is reduced with sodium dithionite. Sodium citrate chelates the reduced iron to keep it in solution. Sodium bicarbonate buffers the solution to pH 7.3 and the method was performed at room temperature. The extracted fluid from a subset of 17 samples from Ridge Well 1 was analyzed using ICP-OES (at Research Analytical Lab, University of Minnesota) to measure the concentration of 27 elements.

CBD extraction was developed to remove Fe oxides, especially amorphous forms, but the method is shown to also remove hydroxy-interlayered aluminum as well as some Al oxides (Shang and Zelazny, 2008). Iron minerals removed by CBD extraction include goethite, hematite, and maghemite, and sub-micron sized magnetite may also be extracted (Hunt et al., 1995). While acknowledging the variety of minerals extracted by CBD method, we refer to these as "extractable oxides" for brevity.

## 4.10 Seismic Multichannel Analysis of Surface Waves (MASW)

Seismic Multichannel Analysis of Surface Waves (MASW) survey was performed across the transect to connect Ridge Well 1 and Interfluve Well 2. MASW survey used a 16 lb sledgehammer as the seismic source in conjunction with a 24 channel Geometrics Geode seismograph and 4.5 Hz geophones. The geophones were spaced 1.5 m apart (34.5 m spread) with a source offset of 10.5 m from the end the geophone spread to avoid near field effects in surface wave generation. At each shot location we stacked five hammer blows to boost the signal to noise ratio. The entire line was then shifted 3 m (twice the geophone spacing) and the process repeated for a total survey line length of 141 m. The seismic data were analyzed using SeisImager/2W[5]. This approach resulted in estimates of seismic shear wave velocity down to ~20 m depth along the transect.

## 5 Results

### 5.1 Morphology and denudation rate

The Manor Series soils are coarse-loamy, micaceous, mesic Typic Dystrudepts that exhibit morphological differences associated with landscape position. Detailed morphology of the Spring Brook soils is described by Fisher et al. (2018). To test the hypothesis that SSA changes with morphological boundaries, we highlight the soils at the ridge and interfluve, which correspond with Wells 1 and 2, respectively.



We identified the C horizon at 84 cm at the ridge by observation of rock structure, silvery gray-green color, undisturbed foliation orientation, and the absence of pedogenic features at this depth within the hand excavated soil pit. At the interfluve
we excavated a soil pit to 124 cm, where we observed yellowish brown coloration, pedogenic mottling, and some disruption of rock fragments, so we did not designate this as the C horizon. A grid of soil recovery probes into the bottom of the soil pit indicated a consistent depth where the color changed to the silvery gray-green bedrock color at approximately 150 cm, which is the depth of the C horizon at the interfluve. In summary, the depth of the soil to weathered rock boundary occurs at 84 cm at the ridge and 150 cm at the interfluve.

**5.2 Mineral specific surface area**

Measured values of total mineral specific surface area (SSA, Figure 3A) reveal a sharp increase in SSA from 3 m to the ground surface (0 m), with total SSA ranging from 9-22 $m^2$ $g^{-1}$ from 0 to 1.5 m, 2-12 $m^2$ $g^{-1}$ from 1.5 to 3 m depth, and 2-5 $m^2$ $g^{-1}$ from 3 to 21 m. Silicate SSA ($SSA_{Si}$) (Figure 3B) increases toward the surface, mirroring total SSA, and reached a maximum value of 10 $m^2$ $g^{-1}$.

Citrate-dithionite extractable oxides ($Fe_d$ SSA=1-($SSA_{si}$/SSA) contribute 36-81% of the SSA found in samples within the top three meters at both landscape positions (Figure 3C), and this SSA is notable because at most only 2.7% of the total sample mass was removed by CBD extraction (Figure 3D). The fraction of SSA contributed by extractable oxides at Ridge Well 1 is 48-81% from 0-3 meters deep and 5-25% below 3 m. At Interfluve Well 2 extractable oxides contribute 36-77% of SSA
from 0-1.5 m, 13-74% from 1.5-3 m and 10-15% below 3 m.



Earth **Surface**
**Dynamics**
Discussions

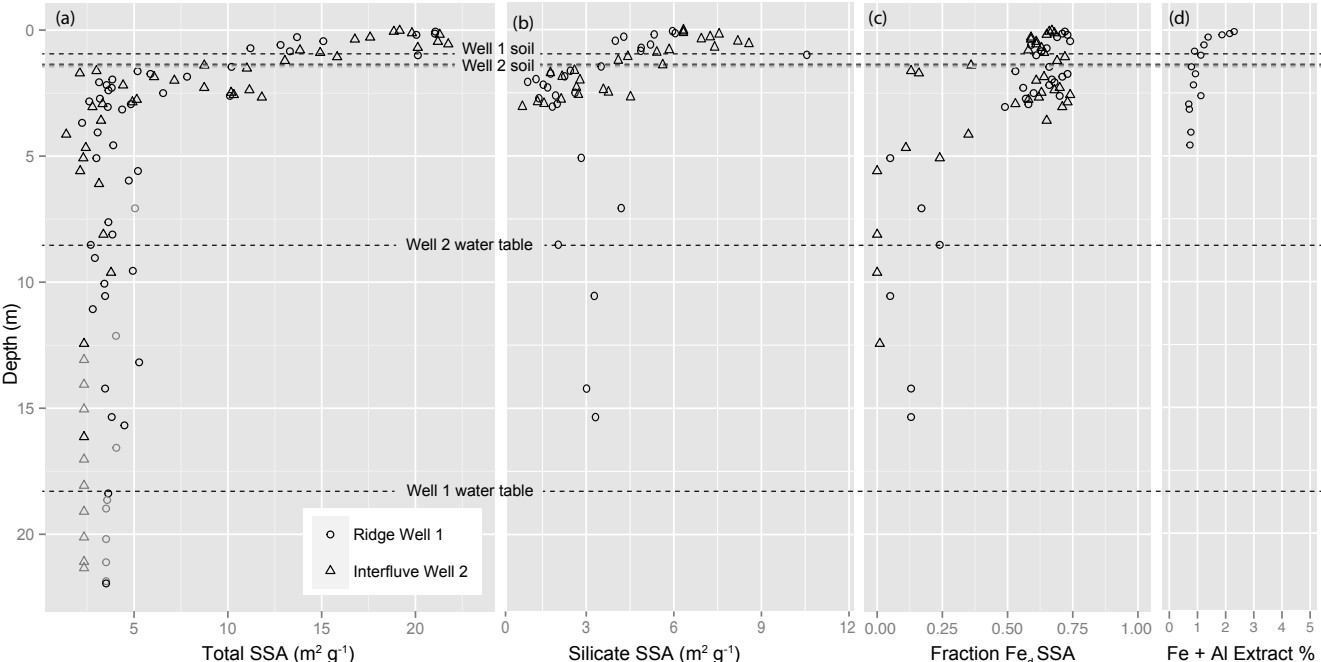

**Figure 3. Total SSA vs depth (a), silicate SSA (b), and fraction of SSA contributed by citrate-dithionite extractable iron (Fe$_d$, C), and silicate SSA (c) reveal that SSA increases toward the ground surface and exhibits an abrupt increase at 3 m. Fe$_d$ SSA reveals** 335 **that most intervals above 3 m contribute over 50% of SSA, and below 3 meters, very little of the SSA is contributed by iron oxide minerals. The percent of Fe and Al extracted (d) reveal that Fe$_d$ SSA, although the majority of the SSA in the top 3 m, comes from a very small percent of the sample. Gray points are interpolated or estimated SSA, no measurements of silicate SSA or Fe$_d$ SSA were made below 3 m in Interfluve Well 2.**


Fe$_d$ SSA increases in direct proportion to SSA$_{Si}$ over the top three meters at both landscape positions as statistically shown in

Figure 4. None of the sharp changes in total SSA, Fe$_d$ SSA, and SSA$_{Si}$ coincide with the transitions from soil to weathered

rock (84 cm & ~150 cm for ridge & interfluve respectively), nor do they correlate with water table fluctuations (Figure 3).



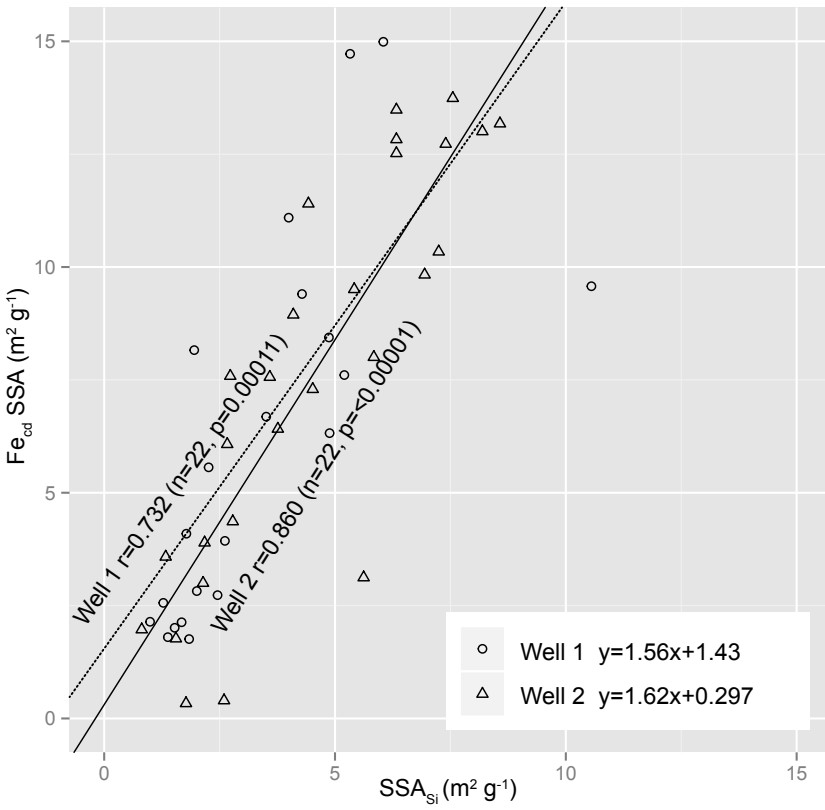

**Figure 4. Correlation of $Fe_d$ SSA with silicate SSA for data from 0-3 meters reveals a strong positive correlation between iron and silicate SSA. The correlation holds true at Ridge Well 1 and Interfluve Well 2.**

The plot of cumulative inventory of total mineral surface area (SAI,

) (Figure 5) reveals where changes occur in mineral surface area production. We applied a segmented linear regression model (Muggeo, 2008) to identify intervals with similar slopes and the breakpoints between these intervals. We iteratively applied segmented regression to optimize fit and break points for each SAI profile, using SAI as the dependent variable in the regression analysis and depth as the independent variable (Figure 5). Ridge Well 1 had two breakpoints at 1.50±0.08 and 3.11±0.07 m (slope 1=$8.71 \times 10^4$ $m^2 m^{-3}$, slope 2=$4.81 \times 10^4$ $m^2 m^{-3}$, slope 3=$7.51 \times 10^3$ $m^2 m^{-3}$, $R^2$=0.9981). Interfluve Well 2 had three breakpoints at 1.02±0.03, 2.22±0.06, and 2.82±0.02 m (slope 1=$1.35 \times 10^5$ $m^2 m^{-3}$, slope 2=$6.07 \times 10^4$ $m^2 m^{-3}$, slope 3=$1.06 \times 10^4$ $m^2 m^{-3}$, slope 4=$5.83 \times 10^3$ $m^2 m^{-3}$, $R^2$=0.9994). The largest relative change in slope within each profile occurred near 3 meters at both landscape positions.





These breakpoints do not correlate well with morphologic features observed from soil pits or drill cores, such as the water

table (~19 m and ~7.5 m for Wells 1 and 2 respectively) or the soil to weathered rock boundary (84 cm and 150 cm for Wells

1 and 2 respectively) (Figure 5). Nor do the breakpoints correlate with the depths of elemental Ca depletion at 10 m in Well

1 and 7 m in Well 2 (Fisher et al., 2017b).

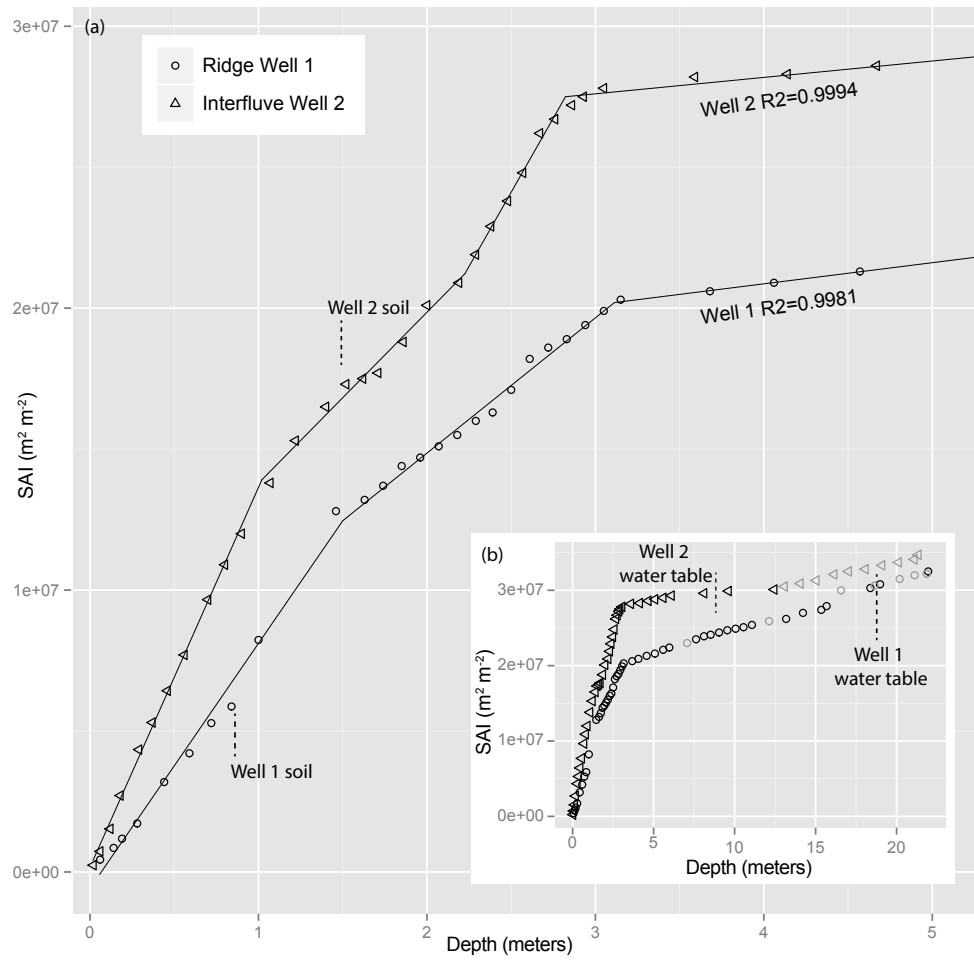


**Figure 5. Surface area inventory (SAI,**

**), reveals trends in the accumulation of mineral surface area. The inset image (B) shows the full depth trend while the larger image (A) reveals the top five meters, where changes in the slope, once combined with erosion rates, may indicate different rates of SSA production. The segmented linear regressions reveal slope changes that do not coincide with morphologically identified**
**boundaries. Morphological boundaries are noted as "Well 1 soil" and "Well 2 soil" indicate the depths where soils transition to weathered rock. Note that distinct change in the slope of SAI, which occurs near 3 meters, does not occur at the lower boundaries of soils or near the water table in either well (B). Gray points in the inset are calculated from interpolated or estimated SSA.**



## 5.3 Mineralogy


Quantitative analysis of bulk XRD reveals that the Laurels Schist is mineralogically variable (Table 1, Figure 6). The primary mineral matrix is composed of quartz, Ca/Na-plagioclase, muscovite, and chlorite group minerals. In Ridge Well 1, despite mineralogical variability, XRD of random mounts of the whole sample indicated a gradual decrease of primary plagioclase group minerals from 7 meters to the surface at the ridge and above 5 meters at the interfluve. The muscovite and chlorite group minerals oscillate with depth in both wells, yielding no clear enrichment or depletion trend.


| Site | Depth (m) | Quartz (%) | Plagioclase (%) | Mica Group (%) | Chlorite Grp (%) | Others (%) |
|------|-----------|-----------|-----------------|----------------|------------------|------------|
| W1 | 0.06-0.14 | 8 | 2 | 50 | 16 | 23 |
| W1 | 1.00-1.46 | 44 | 2 | 21 | 8 | 25 |
| W1 | 2.18-2.29 | 43 | 6 | 25 | 4 | 22 |
| W1 | 3.56-3.68 | 46 | 8 | 17 | 8 | 21 |
| W1 | 4.98-5.08 | 39 | 7 | 23 | 8 | 22 |
| W1 | 6.95-7.07 | 45 | 12 | 12 | 9 | 22 |
| W1 | 8.42-8.52 | 37 | 5 | 27 | 9 | 21 |
| W1 | 10.49-10.55 | 64 | 9 | 1 | 1 | 24 |
| W1 | 12.00-12.13 | 42 | 15 | 15 | 7 | 21 |
| W1 | 13.97-14.22 | 40 | 11 | 14 | 11 | 24 |
| W1 | 15.24-15.35 | 56 | 7 | 10 | 5 | 22 |
| W1 | 18.91-18.98 | 41 | 15 | 14 | 10 | 22 |
| W1 | 21.86-21.95 | 52 | 13 | 8 | 5 | 22 |
| W2 | 0.13-0.25 | 48 | 3 | 12 | 3 | 34 |
| W2 | 0.64-0.76 | 38 | 3 | 14 | 4 | 41 |
| W2 | 1.14-1.27 | 30 | 6 | 25 | 5 | 33 |
| W2 | 2.93-3.05 | 41 | 3 | 12 | 5 | 39 |
| W2 | 4.57-4.67 | 49 | 9 | 8 | 4 | 29 |
| W2 | 5.99-6.10 | 33 | 5 | 16 | 11 | 35 |
| W2 | 7.01-7.25 | 48 | 3 | 2 | 2 | 44 |
| W2 | 9.55-9.62 | 39 | 5 | 13 | 9 | 34 |
| W2 | 10.89-11.05 | 25 | 6 | 25 | 10 | 35 |
| W2 | 13.87-14.06 | 46 | 4 | 11 | 5 | 34 |
| W2 | 15.78-16.14 | 37 | 7 | 12 | 6 | 38 |
| W2 | 17.93-18.07 | 35 | 9 | 9 | 7 | 40 |
| W2 | 21.09-21.34 | 36 | 11 | 16 | 8 | 29 |

**Table 1. Percent distribution of the four major mineral groups of rock samples from the Laurels Schist Formation illustrates how mineralogy is variable with depth in both wells, lacking distinctive weathering trends. Other minerals include may include**
**magnetite, hematite, goethite, epidote, garnet, zircon, and others that are in quantities too low for XRD detection or in crystal forms that are amorphous to XRD.**





**Figure 6. Representative XRD spectra for both wells. Bulk XRD is analyzed on random mounts of pulverized sample with a zincite internal standard (peaks in Å/2θ: 2.476/35.65(1), 2.816/31.35(0.71), 2.602/33.93(0.56)). Clay XRD is performed on oriented suspension mounts of pre-treated sample (see section** Error! Reference source not found.**). Mineral groups are generalized and abbreviated as C: chlorite, I: illite, K: kaolinite, M: mica, P: plagioclase, Q: quartz, V: vermiculite.**

XRD spectra from both sample sites are shown in Figure 6. In the clay XRD spectra we see evidence for chlorite (14 and 7Å), illite (10Å), vermiculite (14Å decreasing to 10Å on heating), and kaolinite (7Å) (Jackson and Barak, 2005; Moore and Reynolds, 1997; Poppe et al., 2001). Where the 7Å clay XRD peak is not greatly reduced upon heating in Ridge Well 1, this suggests that very little kaolinite is present (Poppe et al., 2001), and the observed 7Å peak is a second order chlorite peak.





The peaks for illite/mica and chlorite/vermiculite increase between 1.5-2 m in the bulk XRD samples in Ridge Well 1, but do not appear to change in quantity through the entire depth of Interfluve Well 2.

XRD analysis reveals both kaolinite and clay sized chlorite from soil pits and push cores in the top 1.5 meters at the ridge and the top 3 meters at the interfluve indicated by the destruction of the 7 Å peak upon heating to 500°C (kaolinite) and the
intensification of the 14 Å peak (chlorite) in the same treatment (Jackson and Barak, 2005). The presence of kaolinite is thus difficult to identify in any rock where primary chlorite group minerals share diagnostic d-spacing.

### 5.4 Magnetic mineralogy

Induced and remanent magnetizations were used to analyze the Fe-based magnetic mineral assemblage within a small subset of Spring Brook samples from the ridge. Data reveal a small quantity of primary magnetite within all Laurels Schist samples.
Overlying soils inherited minute quantities of this magnetite from the parent rock, but also display fine grained magnetite (<300 nm), likely pedogenic, enriched in near surface samples. Of the bulk soil, up to 16% goethite was detected in near surface samples, while the same intervals contain <0.5% magnetite, and a similarly small quantity of hematite. Ilmenite was also identified in samples of both the Laurels Schist and its overlying soil. Thus, the vast majority of the Fe-based mineral assemblage in the soil appears to be poorly crystalline goethite, which is consistent with field observations of yellowish
colors within the soil.

### 5.5 Seismic Multichannel Analysis of Surface Waves (MASW)

Seismic Multichannel Analysis of Surface Waves (MASW) survey revealed a gradual, uniform increase in seismic shear wave velocity with increasing depth, from 160 m s$^{-1}$ at the surface to 500 m s$^{-1}$ at 14 m at the ridge Well 1 (Figure 7). The depth to 500 m s$^{-1}$ velocities became increasingly shallow along the transect from Well 1 to Well 2 to approximately 10 m
depth (Figure S2). Shear wave velocities of 500 m s$^{-1}$ are interpreted as soft rock and are not increased by saturation (Lowrie, 1997). The soft rock velocities seem to conflict with rock chip densities, which average around 2700 kg m$^{-3}$ in both wells. Foliation in the Laurels Schist results in discontinuous rock, which effectively slows seismic wave propagation, resulting in low velocities despite the high density rock chips (Clarke and Burbank, 2010, 2011). MASW surveys do not reveal any sharp transitions at ~3 m depth or any of the other depths where we observed changes in mineral SSA (Figure 3 and Figure 5). On
the contrary, MASW results clearly indicate an absence of abrupt change in rock properties over the entire 20 m depth of the seismic cross section.



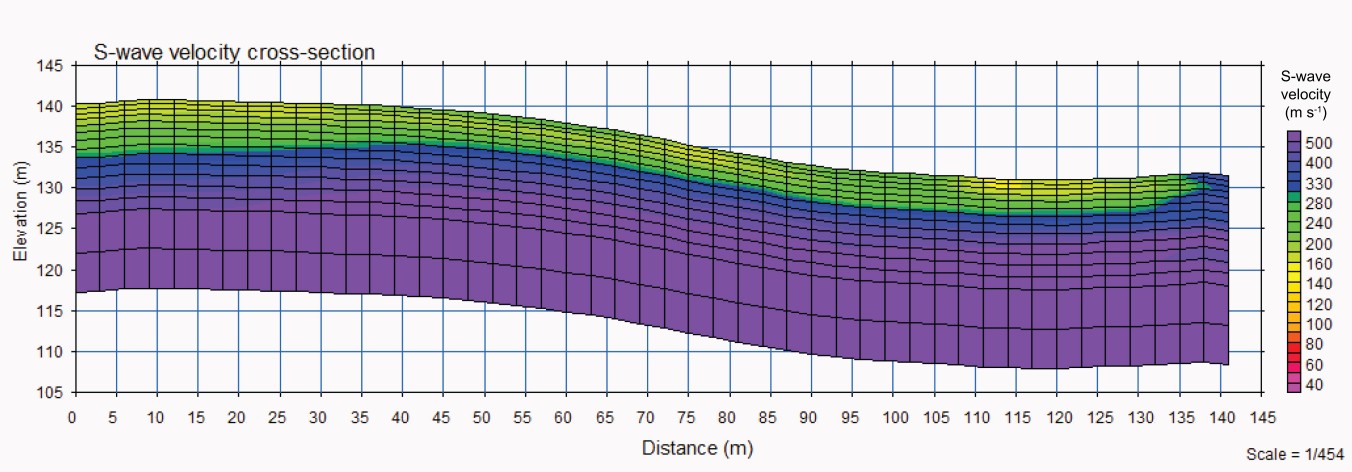

**Figure 7. MASW cross section from Well 1 on the left to Well 2 on the right revealing a gradual change in rock properties from the ground surface, with no subsurface structures detected. The S-wave velocities reach a maximum at 500 m s⁻¹ to a depth of 14 m deep at Well 1, which tapers to 10 m deep at Well 2.**

## 6 Discussion

### 6.1 Specific surface area and rock weathering

Many weathering studies seek to identify the depth to unweathered bedrock, with the understanding that nearly all shallow geochemical and geomorphic processes are linked to underground weathering and hydrology (Carson and Kirkby, 1972; DiBiase et al., 2018; Fisher et al., 2017b; Flinchum et al., 2018; Heimsath et al., 2009; Holbrook et al., 2019; Neely and DiBiase, 2020; Rempe and Dietrich, 2014). Weathering models expect to find an abrupt change in rock properties and identify a rather specific depth to bedrock or the extent of weathering fronts. These weathering models work when the bedrock is relatively homogeneous, especially hydraulically. But where bedrock is heterogeneous, highly fractured, or foliated, such as in sedimentary and metasedimentary lithologies, which cover more than 90% of the Earth's continents (Amiotte Suchet et al., 2003; Blatt and Jones, 1975), finding an abrupt depth to unweathered bedrock can be nebulous. In cases where secondary fractures dominate the porosity, weathering occurs along fractures (or foliations) where water is able to readily interact with bedrock (Gu et al., 2020; Holbrook et al., 2019).

This study of foliated, metasedimentary bedrock revealed no abrupt boundary between weathered and unweathered bedrock. Seismic (MASW) data indicated no abrupt change in rock properties over the entire 20 m depth of the seismic cross section (Figure 7). Elemental analysis of the drill cores indicate removal of Ca and Na to 10 m at Well 1 and 7 m at Well 2 (Fisher et





al., 2017b). The seismic and elemental results demonstrate typical landscape-scale weathering where we expect the influence

of weathering to extend deeper into bedrock along ridges than in lower landscape positions.

**6.2 Processes that promote weathering to the 3 m deep SSA boundary**

We observed an abrupt change in SSA at 3 m deep that records the formation of secondary minerals in the weathering profile. Below 3 m, the weathering profile recorded elemental removal of Ca and Na, which did not generate substantial SSA change. The 3 m depth ($3.11\pm0.07$ m in Well 1 and at $2.82\pm0.02$ m in Well 2) is below the soil to weathered rock boundary

and substantially above the water table. Tree rooting depth and periglacial frost damage are processes that occurred on this landscape that could have influenced a relatively uniform depth on the order of 3 m across the landscape.

The uniform depth of SSA developed at both landscape positions may be explained by the forest cover and tree root influence throughout the study site. Physical disruption caused by rooting and the rootwads of overturned tress (Anderson,

2019; Roering et al., 2010) facilitates water storage and penetration of $O_2$, giving biogeochemical processes access to fresh minerals (Drever, 1994; Hasenmueller et al., 2017; McCormick et al., 2021). The weathering influence by vascular plants can generate abrupt weathering boundaries evidenced by removal of minerals such as plagioclase (Cochran and Berner, 1996). Several of the tree species present in the Laurels Preserve are documented to root to at least 3 meters deep (including Q. rubra, Q. alba, L. tulipifera, C. cordiformis) (Stone and Kalisz, 1991), and tree-throw pit-mound topographic features

(Roering et al., 2010) are common in Lidar data in the study catchment (Figure 2, Lidar DOI: 10.5069/G9T43R00). Physical weathering and chemical dissolution of silicate minerals are closely associated with vascular plants in the rhizosphere (Alexandre et al., 1997; Andrews and Schlesinger, 2001; Berner, 1992; Berner and Cochran, 1998; Cochran and Berner, 1996; Drever, 1994; Moulton, 2000; Pawlik et al., 2016). Physiochemical weathering in deeper zones and in the microenvironments surrounding roots can be further promoted by root and rhizosphere respiration of $CO_2$ and acidic

secretions (Andrews and Schlesinger, 2001; Berner and Cochran, 1998; Cochran and Berner, 1996; Moulton, 2000; Pawlik et al., 2016). These observations suggest that tree roots may have contributed to the observed depth profiles of mineral SSA.

Prior to tree growth, the study site was on the edge of the last glacial maximum, which may have primed the soils for deep rooting by trees and other vegetation. Geomorphologists and pedologists have long been aware that periglacial environments

during the last glacial maximum left a wide range of geomorphic legacy in the continental US that includes the Pennsylvania piedmont (Ciolkosz et al., 1986; Ciolkosz and Waltman, 1995; Clark and Ciolkosz, 1988; Gardner et al., 1991; Hancock and Kirwan, 2007). Frost damage during periglacial conditions can generate a fracture and macropore network (Anderson et al., 2013; Anderson, 2019; Marshall et al., 2021; Marshall and Roering, 2014). Modeling studies based on frost processes throughout the world reveal that frost damage of bedrock can extend as much as four meters deep where the frost cracking

temperature window occurs between -3°C to -8°C (Andersen et al., 2015; Anderson et al., 2013; Anderson, 2019; Hales and Roering, 2007). Anderson et al. (2013) understand frost damage and mobile regolith transport processes as climate driven





processes. Frost damage processes effectively generate conduits for atmospheric oxygen, water, and soil $CO_2$ and facilitate root penetration to the 3 m depth extent where we observe the changes in SSA production.

### 6.3 Secondary mineral formation and SSA generation

We observe an increase in $SSA_{Si}$ from 1-3 $m^2$ $g^{-1}$ below 3 m depth, and from 4-10 $m^2$ $g^{-1}$ from 3 m to the ground surface at both landscape positions (Figure 3). The increase in $SSA_{Si}$ in Ridge Well 1 is supported by the increase in illite and vermiculite (Figure 6). Illite in natural settings contains on the order of 17-41 $m^2g^{-1}$ of SSA (Dogan et al., 2007; Macht et al., 2011) and vermiculite has published values in the same range (Kalinowski and Schweda, 2007; Thomas and Bohor, 1969). The observed decrease of primary plagioclase (Table 1), which weathers to kaolinite, can increase SSA from 1 $m^2g^{-1}$ to 10-

20 $m^2g^{-1}$ (Essington, 2003). Kaolinite formation is favored by acidic environments at the expense of the formation of smectite phases (Jackson, 1963; Jackson et al., 1948), and smectite is absent in the studied samples.

The increase in $SSA_{Si}$ in Interfluve Well 2 does not appear to be the result of increased illite and vermiculite. We also do not observe an increase in chlorite peak intensity in bulk XRD, but the disaggregation of primary chlorite through the weathering profile will also increase $SSA_{Si}$ without impacting the chlorite XRD peaks. At Well 2 the depositional morphology will also contribute to the higher total SSA due to greater fine (< 2mm) material.

$Fe_d$ SSA increased in direct proportion to $SSA_{Si}$ over the top three meters of the drill cores (Figure 4). $Fe_d$ SSA reached 15 $m^2g^{-1}$ at the ridge and 14 $m^2g^{-1}$ at the interfluve, whereas $SSA_{Si}$, the combined SSA of primary minerals and secondary phyllosilicates, was responsible for up to 10 $m^2g^{-1}$ of SSA at the ridge and 9 $m^2g^{-1}$ at the interfluve (Figure 3C). Immediately below this 3-meter boundary the $Fe_d$ contribution to SSA is nearly nonexistent. Yellow-brown coloration is visible in core samples from 0-3 meters at both landscape positions as evidence of the presence of iron oxides, particularly goethite, in these

samples (Figure S1 and Dataset S1).

The contribution of crystalline and amorphous secondary iron and aluminum oxide minerals (citrate-dithionite extractable or $Fe_d$ for brevity) to total SSA ranged between 36-81% of the total SSA over the top 3 m at both landscape positions. Especially significant to this finding is that the extraction removed only 0.8-2.7% of the total sample mass at each interval

(Figure 3D), meaning that the vast majority of the SSA came from a <3% of each sample. Of the extracted phases, 40-75% were iron minerals, primarily goethite (16%), but some intervals include minor contributions from fine-grained magnetite (<300 nm), hematite, and ilmenite. Notably, the XRD did not identify the goethite, even though it comprised 16% of the sample by mass, or the other iron minerals, which is congruent with the small and poorly ordered crystallinity of secondary soil minerals. Iron oxides are known to possess high SSA (Eusterhues et al., 2005; Keil and Mayer, 2014; Thompson et al.,



2011), but that such a small fraction of the sample could dominate the total SSA is meaningful, especially when noting that these minerals were beyond the detection capacity of XRD.

### 6.4 $O_2$ and $CO_2$ as weathering agents

In general, minerals containing iron are weathered by oxidation reactions, while the weathering of silicate minerals is enhanced by acidity. Two bioactive gases, $CO_2$ and $O_2$, are critical reagents for the acid-base and redox reactions of chemical
weathering (Bazilevskaya et al., 2014; Brantley et al., 2013b; Buss et al., 2008; Pawlik et al., 2016; Richter and Billings, 2015). We observe nested weathering fronts driven by different reaction processes (Brantley et al., 2013a; Pedrazas et al., 2021). A silicate weathering front occurred at 7-10 m, where acid-base reactions dissolve Ca and Na and degrade primary silicates (Fisher et al., 2017b), resulting in no secondary mineral formation and little to no change in SSA. A shallower weathering front occurred at 3 m at both landscape positions, where oxidation of iron-bearing primary minerals resulted in
the formation of secondary iron oxides and phyllosilicates. Secondary mineral formation of oxides is a process we expected to be controlled by the presence of oxygen, but the formation of secondary silicate minerals did not occur at this site at depths below 3 m, where acidity was the sole weathering agent.

In most soils, $CO_2$ is produced by soil organisms as $O_2$ is consumed, so the depth profile of $O_2$ is inversely proportional to
pCO$_2$ (Kim et al., 2017; Liptzin et al., 2011; Richardson et al., 2013). Oxygen enables abiotic and microbially-mediated iron oxide formation and is a fundamental process for fracturing rock and promoting silicate weathering (Anderson et al., 2002; Bazilevskaya et al., 2014; Brantley et al., 2013b; Buss et al., 2008; Kim et al., 2017; Maxbauer et al., 2016; Stinchcomb et al., 2018).

Increased soil $CO_2$ generates acidity, which weathers silicate minerals. A 7-10x increase in the partial pressure of $CO_2$ (pCO$_2$) results in a proportional increase in carbonic acid and a 1 pH unit decrease in soil pH (Hasenmueller et al., 2015). Davidson and Trumbore (1995) observed pCO$_2$ to steadily increase to ~70,000 $\mu$L L$^{-1}$ at 8 meters in the eastern Amazon, which is ~175 times more concentrated than $CO_2$ in the atmosphere (~400 $\mu$L L$^{-1}$). Richter and Billings (2015) measured pCO$_2$ ranging from ~12,000 to ~44,000 $\mu$L L$^{-1}$ (30-110 times the atmosphere) at 5.5 meters in the Calhoun Experimental
Forest in the South Carolina Piedmont. Landscape position can more significantly influence pCO$_2$ than depth, particularly where soil moisture is highly variable within a landscape, yet soil pCO$_2$ is invariably higher than atmospheric values (Hasenmueller et al., 2015). Gas measurements in diabase in Pennsylvania record pCO$_2$ ranging over the seasons between ~40,000-80,000 $\mu$L L$^{-1}$ at 4 m deep and in a Virginia granite pCO$_2$ reached its peak at 6 m, where measurements ranged from ~30,000-100,000 $\mu$L L$^{-1}$ through the seasons (Kim et al., 2017). Soil $CO_2$ is high and increases with depth to concentrations
orders of magnitude higher than atmospheric pCO$_2$, and processes that generate $CO_2$ can extend deep into weathering fronts, indicating that acidity from elevated $CO_2$ does not likely limit chemical weathering at depth.



The transition from negligible $Fe_d$ SSA to 36-81% of total SSA at 3 meters deep records an abrupt change in oxidation reactions, which, in the absence of pore gas measurements, is our best way to estimate of the depth of the $O_2$ gas ventilation
front (Anderson et al., 2002; Brantley et al., 2013b; Kim et al., 2017; Maxbauer et al., 2016; Stinchcomb et al., 2018). Secondary iron oxide and oxyhydroxide minerals are often used a geological recorder of oxidation processes and subsurface $O_2$ gas (Anderson et al., 2002; Brantley et al., 2013b; Kim et al., 2017; Maxbauer et al., 2016; Stinchcomb et al., 2018). In steep forested Oregon Coastal Range, greywacke containing pyrite was pervasively oxidized to the depth of 4.5 meters at the ridge top, above which pyrite was no longer present (Anderson et al., 2002). At Shale Hills Critical Zone Observatory in
Pennsylvania, pyrite oxidation extended as deep as 23 meters (Brantley et al., 2013a). Oxidation of iron within biotite in quartz diorite was identified as a source of fracturing and the initiation of weathering in a mountainous landscape in Puerto Rico (Buss et al., 2008). At a granite ridge in the Piedmont, Bazilevskaya et al., (2014) found that the depth of the lower boundary of weathering is controlled by biotite oxidation, which generates porosity and accelerates other chemical weathering processes by facilitating advective transport of water and oxygen. Maxbauer et al. (2016) identified a 25 m deep
oxidation front in sediments in Wyoming.

The oxidative chemical weathering of iron-bearing silicates provides insight into how the oxidation of an iron-bearing mineral generates SSA. Iron-bearing micas and chlorites, for example, oxidize to vermiculite or illite plus an accessory iron oxide or oxyhydroxide (Jackson and Hseung, 1952; Jackson et al., 1948; Righi et al., 1993; Ross and Kodama, 1976). Figure
4 illustrates the strong, positive correlation between $Fe_d$ SSA and $SSA_{si}$, and the correlation plot reveals that the trend is similar both landscape positions. (2 offers an example stoichiometry of oxidation of an iron-bearing mica that is possible at our study site, where Fe is 5-16% of rock mass (Fisher et al., 2018). Note that these reactions cannot proceed without $O_2$ and they generate acidity. Oxygen, not acidity, is the limiting reagent in this type of reaction because soil $CO_2$ is produced at the expense of $O_2$ resulting in decreasing availability of $O_2$ with increasing depth in weathering profiles (Kim et al., 2017;
Liptzin et al., 2011; Richardson et al., 2013).

Iron-bearing mica (3-40 m²g⁻¹) $\qquad$ (2)

$$K(Mg_2Fe^{II})(Si_3Al)O_{10}(OH)_2 + 0.2H_4SiO_4 + 0.25O_2 + 01.1H_2O + 1.25Mg^{2+}$$

Vermiculite (17-41 m²g⁻¹) $\qquad$ Gibbsite (600 m²g⁻¹) $\qquad$ Goethite (60-200 m²g⁻¹)

$$\rightarrow Mg_{0.35}(Mg_{2.9}Fe^{II}_{0.1})(Si_{3.2}Al_{0.8})O_{10}(OH)_2 + 0.2Al(OH)_3 + 0.9FeOOH + 1.5H^+ + K^+$$
$$+ 0.4H_2O$$



| Mineral | SSA of mineral ($m^2g^{-1}$) | Abundance of minerals at 2.5 m | SSA of bulk rock at 2.5 m ($m^2g^{-1}$) | Abundance of minerals at 1 m | SSA of bulk soil at 1 m ($m^2g^{-1}$) |
|---|---|---|---|---|---|
| Quartz[*] | 3 | 50% | 1.5 | 50% | 1.5 |
| Plagioclase[* **] | 3 | 7% | 0.21 | 1% | 0.03 |
| Chlorite[†] | 10 | 13% | 1.3 | 10% | 1 |
| Mica[*] | 10 | 30% | 3 | 20% | 2 |
| Vermiculite[‡] | 40 | | | 16% | 6.4 |
| Goethite+Gibbsite[§] | 100 | | | 3% | 3 |
| Totals | | | 6 | | 14 |


**Table 2. Hypothetical calculations of SSA that correlate with two depths in the Spring Brook weathering profiles using an estimated SSA for each mineral and mineral abundance used to calculate SSA. [*](Essington, 2003), [†](Dogan et al., 2007; Macht et al., 2011), [‡](Kalinowski and Schweda, 2007; Thomas and Bohor, 1969), [§](Borggaard, 1983; Essington, 2003), [**](Brantley and Mellott, 2000).**


Our samples had a maximum of 2.7% or extractable oxides generating more than half of the SSA at many depths in the top 3 m of the weathering profile. We used the reaction in Eq. 2 to model the potential SSA. Assuming a mineral matrix comparable to our measurements in Ridge Well 1, Table 2 shows estimated SSA for each mineral identified. If we consider

the minerals at 2.5 m and assume no oxidation reaction, the SSA of quartz, plagioclase, chlorite, and mica totals 6 $m^2g^{-1}$. If we apply a weathering reaction that oxidizes one third of the mica minerals to vermiculite, gibbsite, and goethite, similar to what we observe at 1 m in Ridge Well 1, we arrive at a total SSA of 14 $m^2g^{-1}$ as shown in Table 2. These simple calculations demonstrate that oxidative weathering reactions similar to Eq. 2 could explain the observed mineralogical and SSA profiles at our study site.

**7 Conclusions**

By applying SSA to assess the extent of weathering in two 21 m profiles at different landscape positions within an unglaciated schist lithology, we discovered that the morphologic boundaries within weathering profiles, namely the soil-weathered rock boundary and the water table, did not coincide with depths where mineral SSA exhibited significant changes. Total SSA, contributed by extractable oxide (Fe$_d$) SSA and silicate SSA$_{Si}$, exhibited abrupt changes at 3 m at both the ridge

and interfluve. The weathering profiles had no other abrupt change in surface area. This finding is supported by mineralogy, rock chip densities, and geophysical survey. SSA measurements provided a better indication of the depth extent of secondary phyllosilicate and iron oxide mineral formation than classic mineralogical methods (this paper) or geochemical mass balance methods on this variable schist bedrock (Fisher et al., 2017b). Mineral SSA is thus a highly sensitive indicator of mineral production from chemical weathering.

Earth **Surface**
Dynamics
Discussions


Measuring SSA enabled us to observe the depth where nested weathering fronts generated new SSA. New SSA emerged at the same depth at both landscape positions (3 m) and new SSA was contributed by oxide minerals and secondary silicates. Due to the initiation of oxide formation at 3 m, this depth indicates the $O_2$ ventilation front. In contrast, weathering due to $CO_2$ and acidity extended deep into the weathering profiles, with a noticeable difference due to landscape position (10 m in

Ridge Well 1 and at 7 m in Interfluve Well 2 (Fisher et al., 2017b)), and did not generate new SSA. Our results indicate that the penetration depth of $O_2$ is a first order control that determined the depth of the secondary mineral production, both oxide and silicate.

Future weathering studies could test the hypothesis that a coupled system where $O_2$ is the limited agent that promotes

oxidation of iron and production of $CO_2$, is a fundamental mechanism that drives the production of both secondary silicate and oxide minerals. We arrive at this conclusion from: (1) our observation of a strong positive correlation between iron oxide ($Fe_d$) SSA and silicate $SSA_{Si}$ in the upper 3 m (Figure 4); (2) the ubiquitous negative correlation of $CO_2$ and $O_2$ as geochemical weathering agents throughout biogeochemical literature (i.e. $O_2$ for iron oxides and carbonic acid from elevated $pCO_2$ for phyllosilicates); (3) the ubiquitous trend of decreasing subsurface gas $O_2$ with depth to concentrations near zero at

the depth where iron oxides appear and reduced iron minerals such as pyrite disappear throughout biogeochemical literature; and (4) the typical trend of increasing subsurface $pCO_2$ with depth to concentrations that are typically 1-2 orders of magnitude higher than in the atmosphere throughout biogeochemical literature.

Measuring SSA is not currently part of the standard suite of measurements applied in weathering studies despite it usefulness

in capturing many of the effects of physical and mineralogical alterations. SSA may be especially useful in highly heterogeneous metasedimentary bedrock, where the variability of silicate minerals makes the mineralogical changes difficult to associate with chemical weathering. Our work shows the utility of SSA measurements in characterizing important chemical weathering processes, which are more typically estimated through laboratory measurements of mass change of rock forming elements and their associated minerals, to understand critical hydro-biogeochemical fronts within weathering

profiles.

**Author contribution**

Beth A. Fisher was the lead author of this article, lead the Rotosonic drilling, soil sampling, specific surface area analysis, bulk density analysis, XRD mineralogy (bulk and clay), and the extraction chemistry. Kyungsoo Yoo was the PhD advisor for Dr. Fisher, and he co-authored the funding efforts for this work, trained and assisted Fisher in data collection, analysis,

and interpretation, and provided significant feedback on the article contents. Anthony K. Aufdenkampe co-authored the funding efforts for this work, co-lead the Rotosonic drilling and soil sampling, and provided substantial contributions to the

manuscript narrative and data interpretation. Edward A. Nater provided extensive training on XRD procedures and assisted with XRD data interpretation. Joshua M. Feinberg provided training on iron mineralogy procedures and data interpretation. Jonathan E. Nyquist graciously volunteered his time and lead a group of students through the acquisition, analysis, and interpretation of the MASW data specifically toward the effort of understanding this weathering profile; and Dr. Nyquist also provided thoughtful review of several versions of this manuscript.

### Acknowledgements

The authors extend gratitude to the National Science Foundation for funding this work through the Christina River Basin Critical Zone Observatory (0724971 and 1331856). We would like to thank Brandywine Conservancy, Dr. Phoebe Fisher, and the field team at Stroud Water Research Center, especially D. Montgomery and S. Hicks for immeasurable assistance with the field component of this project. We thank Dr. Nicolas Jelinski for assistance with laser particle size analysis. We wish to thank Dr. Michael Manno and the UMN Characterization Facility for training and assistance with x-ray diffraction. Dr. Fisher would like to express appreciation for abundant laboratory assistance of M. Behling Roser, J. Davis, and A. Durr.

### Data availability

All data in this publication are available at the University of Minnesota Digital Conservancy:
Fisher, B. A.: Geomorphic controls on mineral weathering, elemental transport, carbon cycling, and production of mineral surface area in a schist bedrock weathering profile, Piedmont Pennsylvania, University of Minnesota. [online] Available from: http://hdl.handle.net/11299/183377, 2016.

### Sample availability

Contact author for sample access. Samples are archived at Stroud Water Research Center (soil), University of Minnesota (drill core), and Minnesota State University, Mankato (pulverized soil and drill cores).

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
