# Peer review of "Mineral surface area in deep weathering profiles reveals the interrelationship of iron oxidation and silicate weathering"

_Earth Surface Dynamics, 2022_

## Author Response (AR1)

**Response to Reviewer #1**

**Note: The reviewer comments are in light text, the authors replies are in bold.**

General comments

This manuscript presents detailed observations of mineral surface area and mineralogy in two deep profiles in a watershed and uses these data to understand processes involved in the development of weathering profiles. The profiles extend through soil and weathered rock to a depth of ~20 meters, and hence are a welcome contribution to understanding incipient stages of weathering and profile development. The authors show that measured specific surface area increases from the bottom to the top of the two profiles. The slope of this relationship (SSA vs. depth) shows a prominent break at a depth of 3 m. This depth coincides with the appearance of secondary minerals, but not with macro-scale morphological boundaries in the profile. Elemental depletion (tau) profiles (presented in an earlier publication) show that elemental depletion occurs to several meters below the 3 m boundary. The authors infer that oxidation weathering reactions play an important role in development of the weathered profile and provide lengthy conjectures on physical processes that also operate to this depth.

I found the surface area data interesting particularly coupled with information on mineralogy and morphology of the weathered profile, although the presentation is difficult to follow. I also found several aspects to the manuscript in need of improvement. It would help to pull together the key relevant observations into one figure-- SSA, element depletion profiles, mineral abundances (at least for key minerals), morphological boundaries. The geomorphic context was not well described; this is relevant to the observations in the site the authors refer to as an "interfluve", where there is apparently an incised colluvial deposit on which the weathered profile is developed, as well as to the big picture of how this landscape is forming. There is insufficient characterization of the weathering processes invoked by the authors. Finally, the authors seem unaware of other work on mineral surface area in weathering. Addressing these issues probably constitutes major revisions, but once handled, the manuscript should be worthy of publication.

**The authors have completed a major revision of the manuscript after careful considerations of all comments and suggestions from reviewers 1 (this reviewer) and 2. Almost all figures have been revised. We believe the manuscript is much improved as a result of these major revisions.**

**Many of the criticisms of Reviewer 1 are overly negative in content and tone and are often contradicted by the reviewers own comments elsewhere in the review or in the papers that they cite. Overall, we refute the reviewers assertions that we are unfamiliar with the literature on mineral surface area, weathering, soil formation, hillslope geomorphic, and periglacial processes, which we support within our revisions and in our responses below.**

Geomorphic context:

The two profiles are located on a ridge and an "interfluve". An interfluve is a drainage divide between two adjacent rivers (Latin fluvius for river). In this manuscript, the region called an

"interfluve" seems to be a gullied colluvium deposit. In the caption to Figure 1, it is described as both convergent and convex. It is unclear whether or how often either gully contains channel flow, but the impression is that neither gully qualifies as a river. Why not call it what it is? Gullied colluvium. There must have been a good reason why the gullied colluvium/interfluve was selected for the effort of coring, but that reasoning does not come out in this manuscript. It is not representative of the rest of the watershed—so the choice must have been made for some other reason.

**USDA defines interfluve as a colluvial deposit that has been incised. Carter & Ciolkosz 1986 and Merritts & Rahnis, 2022 both describe how periglacial frost processes during the LGM in central and southeastern Pennsylvania have formed characteristic colluvial landscape features through solifluction creep that are prolific in the study area. Dr. Merritts visited the study site during Amtrak Club field tour in May 2015 and identified numerous examples of periglacial colluvium at the study site and confirmed our characterization of the area around Well 2 as an interfluve formed by post-colonial channel erosion and incision of a periglacial colluvial deposit.**

**Our "interfluve" designation is therefore supported directly by Merritts in 2015 and also by several publications on these periglacial features in the Mid-Atlantic. This definition is common throughout geography textbooks, but it has clearly been a struggle for the weathering community to accept this interdisciplinary explanation, which is why the text included a definition and discussion.**

**The authors have noted the repeated use of the reviewer of the word "apparent" in every reference to "interfluve" and our observations of its surface and subsurface expression. We interpret the reviewers choice of this word as an effort to continual cast doubt on our use of that name for this geomorphic feature. The authors drilled, dug, described, and analyzed the soil and rock, and consulted the literature and many geomorphic experts. We stand by our identification of this landscape feature, and hope that we might inform the weather community of its existence and formation processes.**

Of interest is the timing and cause of the apparent transition from accumulation of colluvium (or perhaps steady state) to incision and how this might affect the weathered profile. The transition from accumulating to eroding should affect the weathered profile in some way. Some consideration of when this occurred seems relevant to understand the profile.

**In the paper we have enhanced our text to more clearly describe that the colluvium was formed after the LGM, about 9 kyr to 15 kyr ago ( Prentice et al., 1991; and supported by Fisher et al 2018, in which we calculated a minimum landscape age of 19 kyr and an erosion rate of 50 m My−1,). We also more clearly describe that incision of the interfluve occurred during deforestation in the early 1800s, about 150 to 200 years ago, and that the present forest is likely about 150 years old.**

Curiously, although the schematic diagram in Figure 1 implies that colluvium is thick at the

gullied colluvium site and absent at the ridge site, there is no evidence of this difference in any of the data shown. What is going on? Why is there no distinction between a weathering profile developed in colluvium and a weathering profile developed in bedrock? This needs explanation.

**The soil morphology data and description published in Fisher et al 2018 make clear distinctions between the ridge top and interfluve soils. We refer to that paper here. Our understanding is that this and most other journals would prefer that we not publish the same data twice.**

The geomorphic context also comes up with respect to the weathering processes considered. There is an extensive conceptual discussion of frost cracking and allusion to nearby glaciers. Where was the glacial margin? When was the Last Glacial Maximum (LGM) in this area? The inferred mean annual temperatures present and past could be used to assess the depth over which frost cracking has occurred (and is occurring). Here is another place where more consideration of the time at which the profile switched from accumulating material (pushing weathered rock/soil, kinks in the surface area profile, soil horizons, etc. down relative to the surface) to the present eroding (gullied) state is relevant.

**LGM was referenced in the text along with MAP and MAT for the eras from Prentice et al., 1991. The authors do not believe it is really a fact that we need to argue or prove that the study site experienced periglacial conditions. We decline to add an additional figure to demonstrate the LGM boundary in PA.**

Another missing aspect of the geomorphic context is a discussion of denudation rates and of regional landscape evolution. The long term denudation rates in an eroding upland (which this site appears to be on the whole) set the timescale for the weathered profile. The regional landscape evolution would describe the proximity to the ice sheet margin, periglacial landscape features, the dynamics of the river into which this catchment flows. In essence, a brief outline of the regional earth surface dynamics.

**Denudation rate from Fisher 2018 has been added to the manuscript.**

Weathering processes

The discussion of weathering processes needs some work. Many weathering models rely on chemical processes in which diffusion limits reaction progress (whether chemical or mechanical —e.g. frost cracking), and so produce weathering profiles that do not have sharp boundaries. The authors seem not to understand this. Fractures are a way to overcome the limitations of diffusion in chemical processes, as water and reactants can be transported along fractures and then the diffusion process is involved only from the fracture surface. This is shown in models of spheroidal weathering (Fletcher et al., 2006) and fracture flow (Pandey & Rajaram, 2016;

Anderson et al, 2019), and in datasets from cores (e.g., Holbrook et al, 2019).

Moreover, the authors assert that frost and tree rooting mechanisms should produce a weathering front at a uniform depth of 3 m across the landscape, but offer little to support this conjecture. In any case, I would expect slope aspect to produce non-uniform depths of frost and tree rooting influences.

**This work on weathering of geochemically variable schist bedrock has not fit the current narrative of weathering models, which is a fundametal finding of this paper. We show multiple lines of evidence for:**

- **weathering that is gradual, not abrupt, so we decline to name a specific depth to saprolite**
- **geochemical and geophysical weathering indicators that do not align with morphological boundaries (e.g. depth to soil C horizon)**
- **The water table was not observed to be linked with any of the reaction fronts. Rempe and Dietrich and a 2010 abstract by Dietrich**
- **no isovolumetrically weathered saprock (Brantley & Lebedeva 2011) and no crumbly and plasticly-cohesive saprolite (graham 2010); instead the bedrock was weathered only between plates of schist, where the rock is foliated. This is based on observation and measurement of rock chips density throughout the profile.**

**The authors are fully aware of the weathering literature and see this review has a demonstration of the difficulty we have had in getting this measurement of SSA of an entire weathering profile through peer review. The weathering community does not seem to know what to do with a full profile of surface area measurements and they do not seem to allow work that does not fit the existing weathering narrative, which seems to apply very well to igneous rock and apparently Pennsylvania shale weathering profiles. On the other hand, the weathering community has very few publications studying weathering processes on more heterogenous sedimentary and metamorphic bedrock, which are the more common bedrock types worldwide. This paper aims to correct that imbalance, and shows data that do not neatly fit into paradigms established for much more homogenous bedrock formations.**

**One common feature of our weathering study with the existing weathering models is that we definitely observe nested weathering fronts (Brantley 2013), and quite possibly expand on this idea by observing elemental depletion as a weathering front that is separate and distinct from secondary mineral formation.**

**Based on our knowledge of the existing weathering models, we were intentional to not use the terms saprock or saprolite because our profile does not contain weathered rock that can be crumbled by hand and is plastic when wet (saprolite per Graham 2010) nor does it contain rock that is fractured, friable, and can be crumbled by hand (e.g. Anand and Paine, 2002). Instead we measured dense plates of solid schist that only reveal weakness in the planes of metamorphic foliation, through our entire weathering profile.**

**The big observation of this work is that SSA enabled us to detect where the nested weathering fronts produced secondary minerals from both acid-base and redox reactions.**

**We also observed that the production of secondary minerals occurred at a separate nested front than elemental depletion. We hope that the weathering community can incorporate these observations into the current weathering models.**

Finally, the discussion of surface area production and weathering processes is a little convoluted. The heart of the matter is that a prominent kink in the SSA profiles occurs at about 3 m depth, which is apparently unrelated to the depth of element depletion (shown unfortunately only in an earlier paper). The authors argue that O2 driven oxidation reactions are driving this mineral surface area production (after having spent a page asserting that frost cracking and tree root growth was responsible for the surface area production). A long argument is put forward that CO2 increases with depth in soil (most likely true), while O2 decreases with depth in soil (also most likely true), although they do not measure these gases. There seems to be a lack of acknowledgement of the effect of CO2 created acidity (which produces a weak acid) compared to O2 driven sulfide oxidation, which produces a strong acid. There is also an abundant literature on the effect of organic acids on silicate weathering rates, none of which is considered. Because the chemical weathering data is presented in an earlier paper, it isn't possible to understand quantitatively how much silicate weathering occurs at various depths, but it seems entirely likely to me that the rate is enhanced where boosted by organic acids produced at shallow depths and strong acids that are by-products of oxidation reactions. The leading edge of weathering does not need to be where the rate is fastest—indeed may not be. I made few line-by-line comments in this section because I think it needs major revision.

**We have more carefully divided out the discussions of each of the topics (periglacial processes, rooting depth, and then O2 and CO2.**

**Section 6.4 has a lot of content on acidity from CO2. The reviewer notes that we did not measure CO2 and O2, yet we followed the well established precedent in the literature of inferring likely CO2 and O2 depth profiles from indirect evidence. We note that these processes are discussed in detail in Brantley 2013, which makes clear that certain secondary  minerals form in the presence of oxygen, and others form due to acidity. The following authors all discuss the dynamics of CO2 and O2 and their weathering products: Bazilevskaya et al., 2014; Brantley et al., 2013b; Buss et al., 2008; Pawlik et al., 2016, and all of them claim higher or lower levels of oxygen, but none of them measured either gas.**

Other work on mineral surface area

The claim that this work was the first ever measurement of total mineral surface area in a weathered profile caught my attention. It is true that I cannot think of another work that presents an integration of measured mineral surface area in a profile. However, the total mineral surface area in a watershed was estimated long ago by Michael Velbel (Velbel, AJS, 1985) and Tomas Paces (Paces, GCA, 1983) in order to compare field (watershed) based weathering rates with lab-based rates. Clow and Drever (Chem Geol, 1996) measured mineral surface area in a "nanocatchment" in their pursuit of the lab-field weathering rate discrepancy. These authors were chasing a different question than the present manuscript, yet their work is relevant and seems

worthy of mentioning. I don't know the surface area literature well, but there is much more to plumb on all the ins and outs of measuring and interpreting surface area data.

**We agree to remove the claim that this paper is the "first ever measurement". The first author has learned a lot about "firsting" in their study of decolonizing science, and choose to remove this statement.**

**We appreciate the suggested publications, but explain below why we did not use them:**

- **Michael Velbel (Velbel, AJS, 1985) ESTIMATED SSA. It's not the same as measuring it.**

- **Tomas Paces (Paces, GCA, 1983) I'm very aware of the mineral weathering rate constants used in geochemical models, where modelers make orders of magnitude adjustments in mineral-specific surface area to tune a model in GWB or PHREEQC. Again, this paper doesn't include SSA measurements, and it's not looking at bulk compositions of rock or soil.**

- **Clow and Drever (Chem Geol, 1996) did measure SSA directly, but did it on mineral separates and on size fractions rather than on the bulk material. They also restricted their measurements to soil, for which there are abundant SSA measurements, such as the many papers by Larry Mayer. All of Clow and Drever's measurements were within the top meter of SOIL, which is not at all comparable to two 20-meter deep weathering profiles.**

I was surprised that Lixin Jin's nanoporosity measurements in a 20-m deep borehole in shale were not cited (Jin et al., Am Mineral, 2011), as they find changes in nanoporosity (and calculated SSA) closely associated with the onset of feldspar dissolution and of dissolution of chlorite and illite. The differences in conclusions should be discussed.'

**We did not cite Jin et al. 2011 because they CALCULATED SSA in the weathering profile. They do not provide SSA measurements that are comparable to our data, although they did measure SSA on some soils in the study site.**

**Furthermore, Jin's 2011 calculated SSA values are exceedingly high (up to 300 m2/g in the soil), which they compared with N2 adsorption SSA from their own 2010 publication where measured SSA was a maximum of 30 m2/g. I see a potential flaw in Jin's SSA calculation method in this order of magnitude difference in measured vs calculated SSA and did not find it within the scope or purpose of our manuscript to address this discrepancy.**

**Jin inferred that oxidation extended to 23 m deep based on a presence of a combination of ferric and ferrous iron. They speak of an ID of ankerite as evidence for this depth of oxidation, but also note that ankerite does not show up in some of the intervals. They never identify which minerals hold the Fe3+ that they found. However, these two forms of iron occur together in several primary mineral structures, and their proper detection and identification depends on magnetic mineralogy, which Jin did not do. In our study, even the**

**primary magentite, which caused the rock chips to stick to a magnet, did not show up by XRD, but was an extremely clear signal by magnetic mineralogy.**

**Jin said: At 5 m depth, an abrupt increase in porosity and [calculated] surface area corresponds with onset of feldspar dissolution in the saprock and is attributed mainly to peri-glacial processes from 15 000 years ago.**

Detailed comments

Figure 1

- It's unclear what is to scale and what is not on this schematic. Given that a lidar DEM is available, why not show an actual topographic cross section?

**The schematic is to scale in both vertical and horizontal  dimensions based on the Lidar DEM, but it is not a direct elevation plot because the features do not align along a simple single line.**

- After the laborious definitions of bedrock and weathered rock in the text, these are ill-defined on the diagram. Soil, carefully defined in the text, is not shown. Colluvium is faintly shown (hard to see), but was not defined in the text.

**Previous commenters requested that the definitions be moved out of the figure caption and into the text because they are "laborious." As a matter a preference, the authors will keep the definitions in the text. We have extended the specification of "gravity-driven colluvial transport" to a more explicit definition: which is unconsolidated material that has been moved by gravity-driven soil creep and other erosion processes.**

- Given that this is a schematic diagram, why show SSA and elemental depletion curves—presumably also schematic-- It would be much more informative to show the curves elsewhere, quantitatively, rather than schematically.

**The figure was made to vertical scale and this has now been noted in the caption.**

- Are the divots in the cross section where the gullies are located?
- **Yes, the caption says so.**
- Is the water table measured, and if so, when? Is the range of water table depth variations known? What are these?
- **Water table measurements and ranges added to Fig 3. We continue to report the mean water table for brevity.**
- The caption describes the interfluve as a "convergent area between two ridges". Interfluves are divergent areas between two rivers.
- **The rivers can be intermittent to qualify as an interfluve.**

Figure 2

- Hillshades are pretty, but not quantitative. Can you add topographic contours? Several topographic features described in the text are not visible at this scale (e.g., 2-3 m circles, tree throw mounds)
- **I do not have more than the digital image of the hillshade and cannot provide contours. At the resolution provided to the journal, the features described are available. Perhaps the print resolution was insufficient in your copy.**
- Too much text on the figure—obscures the topography.
- **Modified**.
- What are black dots, white circle, white triangle? Use a key.
- **modified.**
- (I would call the area around the number 6 in the label "Toeslope Pit 6" an interfluve, by the way.)
- **Pit 6 is toeslope at the edge of an eroded gulley. It's not an interfluve, but the reviewer's comment here clarifies for the authors some of the confusion of the reviewer on this topic. This comment reinforces the need to define the geomorphic positions for the reader, which is why we presented the hillslope schematic, including an interfluve, in Figure 1. Perhaps the NRCS The Field Book for Describing and Sampling Soils, version 3.0 (Schoeneberger, Wysocki, Benham, and Soil Survey Staff, 2012) would help guide the reviewer in how we assigned these landscape positions. I have added this citation to the manuscript.**
-
- Show the seismic line, and where the topographic profile you can make for Figure 1 are located.
- **noted**.
- Please identify Spring Brook, the spring at its head, and the "historic gully".
- **added**.

Figure 3

- This figure might be more effective if it was annotated in a way that the 3 m depth were illustrated (perhaps on the vertical scale).
- **Noted.**
- Identify the base of soil, saprolite, and weathered rock. It's unclear why water table is featured on these figures so much more prominently than the weathering boundaries. You may need separate plots for each profile.
- **Soil is already marked. The authors decline to label saprolite and given the gradual transition from weathered rock to unweathered rock, we also decline to mark this as a line on the figure.**
- It would be useful to see a bulk density and grain density profile as well.
- **Bulk density is calculated. Not sure what you mean by grain density.**
- How are SSA measurements interpolated or estimated?
- **Linear is added.**

Figure 5

- Why not plot with depth on the vertical axis (as in Figure 3)?
- **We chose this to focus on the values of SAI.**
- Show soil, saprolite, weathered rock boundaries
- **See previous response to this request.**

Figure 6

- Opps! Error in the caption
- **Users of British English prefer analyse, while American English users have standardized around analyze.**

Figure 7

- Please show the locations of the wells on this cross section (and show where this transect is located on Figure 2)
- **Noted**.

Line by line notes

L11     Define "mineral specific surface area".  It's unclear what the surface area is being normalized to—whether it is each mineral (as mineral specific implies), or the usual surface area per unit mass of material.

**It is now defined. We specify that it is specific surface area (mass normalized) of all minerals in the bulk material to emphasize that the structures of minerals are what contribute to SSA. We've never observed an author being requested to define this common usage.**

L11     Suggest "increases" rather than "is generated"

**Modified**.

L15     What is an SSA profile?

**we describe it as 21 meter deep. additional words would be redundant here.**

L17     soil to weathered rock is a single boundary, so change boundaries to boundary, or better yet, say "soil to weathered rock boundary".

**modified**

L18     Are there multiple SSA boundaries at 3 meters?  I'm confused.

**modified.**

L19     7 and 10 meters *depth*

*modified.*

L43-44 The definition of weathered rock limits weathered rock to chemically altered rock. Often, physical weathering processes precede chemical weathering processes (see work by Fletcher,

Brantley & Buss; Goodfellow).

**modified**

L46    isovolumetric***ally***

***corrected.***

L65-67 This sentence is problematic. Is the conversion of weathered rock to soil termed "soil production" only where the chemical weathering rate in the soil is proportional to this rate (as the sentence states)?  What is the process called where the chemical weathering rate is not proportional to the soil production rate?

Moreover, most of the works cited here to support this contention do not measure both chemical weathering rate in the soil and soil production rate.  Gilbert (1909) was only concerned with the detachment of particles from rock into soil, his "soil production", and does not present measurements of either this rate or chemical weathering rates. Heimsath et al. (1997) used cosmogenic radionuclides and an assumption of steady state (also assumed by Gilbert) to determine soil production rate; they did not determine chemical weathering rates. Raymo and Ruddiman (1992) is about uplift driven climate change; they do not quantify soil production rates or chemical weathering rates (although they do discuss proxies for the latter, such as the Sr isotope record in seawater).  Riebe et al (2004) do measure both soil production rates and chemical weathering rates, but do not find that they are "directly proportional", as claimed here. West et al. (2005) looked at erosion rates and silicate weathering rates, but never mention "soil production".

**The authors do not see the notion of soil production being proportional to chemical weathering as a controversial statement at all. But the sentences have been removed because they are not a critical point in the manuscript.**

L82    In this sentence, "weathering fronts" are limited to features associated with chemical processes, another instance of ignoring the role of physical processes. Frost cracking is not a chemical process.

**Chemical is removed. Most often authors define fronts by chemical properties such as elemental or mineral removal, but it's true that physical processes are part of weathering. Given that this paper's most significant "front" is identified by a physical measurement, we appreciate the note to remove the chemical designation here. We note the assertion that the authors are ignoring the role of physical processes as untrue and lacking in decorum and we see no changes to be made for that comment.**

L95-100  I often find it artificial to state a hypothesis for a field-based study, and instead find it more useful to outline a broad inquery. Rather than "We hypothesize that x will result in y", it's more genuine to state something like " We pursued defining the relationship between porosity,

mineral surface area, and weathering processes". The attempt to write a hypothesis for this study —an effort surely done after the measurements were made—led to the inclusion of an observational detail about weathering front depths in this particular landscape in a section titled "Hypothesis". The end result is a very local hypothesis that does not engender generality.

**We did hypothesis driven research. The entire field study was designed around these hypotheses as we designed the field installations and drilling effort. Once again the authors note the lack of decorum in the comment that this was "surely done after the measurements were made."**

L105    Much of the text in here repeats verbatim the figure caption. Choose one place to describe the oddity of the incised gully site and do not cut-paste it elsewhere.

**Caption was modified.**

L123    Foliation is an alignment of minerals or planes of weakness and so cannot weather.

**clarified.**

L124    platy *fragments* of rock (?)

**Fragments suggest it's broken.**

L130    Strictly speaking, it is streams rather than watersheds that are ordered.

**Watersheds are commonly described by the order of stream that flows out of them. This site description is in multiple peer-reviewed publications.**

L131-132  Confusing description.  Is the "historic gully" one of the gullies that bounds the "interfluve"? Neither Spring Brook nor the historic gully are identified on Figure 1 or Figure 2. There appears to be a branch in the incised forms on the hillshade in Figure 2, with a confluence that lies between the points labeled Swale Pit 4 and Interfluve Pit 5. Is the historic gully one of these branches? Perhaps, but that does not fit the description of a discontinuous gully segment located upgradient from Spring Brook.

**Figure has been modified with labels.**

It's also confusing to read that the historic gully is a depositional swale.  If the historic gully is in the deep colluvium shown in Figure 1, then the geomorphic history is yet more complicated: 1) accumulation of deep colluvium in a hollow or swale, followed by 2) incision of a gully or two gullies (?), followed by 3) erosion (probably diffusive processes) of a convexity between the gullies, followed by 4) development of an A horizon in the eroded colluvium that is apparently indistinguishable from the A horizon found at the ridge site.  Writing this history raises several points of ambiguity. The historic gully is described as "up gradient" of Spring Brook, a spring fed perennial stream. The cross section (Figure 1) and the hillshade map (Figure 2) do not show an incised gully that is up gradient from a channel head. Is the a-horizon found all the way across

the interfluve, or is it incised (adding yet another period of stasis followed by incision).

**Modified with a clearer timeline and figure changes should clarify the other issues brought up in this comment.**

L137    2-3 m diameter circular features are not readily visible on Figure 2.

**They are now labeled.**

L141    The *present* climate of Spring Brook…

**modified**

L144    I don't think ybp is in standard usage; when used, bp is capitalized. Follow journal standard.  I would use 18-12 ka here, and 9 ka a few lines down, since these are specific dates, rather than timespans.

**Modified**.

L145    …which represents a *periglacial climate at the Last Glacial Maximum*.

**modified**

L164    How is a core sieved?

**L157-9 says, "While the cores were partially pulverized by the Rotosonic drilling action…." the partial pulverization of the drilling method, which was applied without drilling fluid to preserve all the sample, resulting in rock chips and some rock powder. Thus we used a standard wire cloth soil sieve with 2 mm openings (known as #10).**

L167    Were soil pits excavated to the top of weathered rock?   Word missing in this sentence— "within a few meters of _______"

**… "the boreholes" has been added.**

L177    I find it odd that fine material created by the grinding action of the drill is the source of samples used in this study. Were any core samples analyzed?

**N2 adsorption SSA measurements included any chips of rock smaller than 2 mm. The content of this section discusses how we compared pulverized rock SSA with cubes of the same particle size class to demonstrate that the SSA we were measuring was from mineral structures and not simply a product of pulverization. We used N2 adsorption for all SSA samples, and the specimen chamber would not allow for rock chips. A study of these rocks by other methods, such as neutron probe, would be welcome.**

**A grad student from Penn State requested a subset of these drill core samples for neutron scattering analysis. This lead to an AGU presentation in 2016. Despite request for**

**continued discussion and the opportunity to contribute to the interpretation and authorship, the authors have not been consulted and do not know the fate of these samples or this work.**

L203    See my general comments above.

**Noted**.

L204    Perhaps use "land" surface area rather than "ground" surface area, given the description of samples being generated by grinding (L177). to

**Noted.**

L218    How were soil cores obtained?

**Soil recovery probe details added.**

L226    What impeded measurement of bulk density over the intervals mentioned, which cover 75% of the profiles?  What is the material? Bedrock and weathered rock over some fraction of it, but I cannot tell where these boundaries are.  Colluvium? Use same approach as for soil. Saprolite? What is the problem?

**As stated in the text, the recovery of rotosonic cores was incomplete at some intervals. It seems like you are expecting specific depths for the bedrock to weathered rock boundaries in each well.**

L229    What mass of rock chips was used for each density measurement? Water displacement is difficult to do well.

**The drill cores broke into pieces along foliations. Masses ranged from 18-390 grams. We did not find this to be a difficult measurement to do WELL, and we did 3-7 replicates of each interval.**

L230    Plastic or wax coating of friable material for measuring density is a much older idea than Jin et al. (2010).

**True, it's usually applied as a soil clod method, but Jin is where we found the suggestion to apply this method to rocs. Citation has been changed to the USDA methods manual.**

L235   Why assume 32% porosity in the rock? This is an enormous value. I expect porosity to increase toward the surface, and to be much higher in soil and colluvium than in saprolite or weathered rock. There is often an abrupt change in bulk density across the soil/weathered rock interface, but this assumption obliterates it.

**Rock chip densities did not change over your expected abrupt change in bulk density. The authors had already addressed these concerns in the text. In short, this decision was made (and explained) that this was a low bulk density for rock, which would provide conservative calculations of inventories. The purpose of bulk density was to be used for the SSA inventory calculations and holding bulk density as a constant fraction of the measured rock chip densities allowed for the rock weathering to reveal itself through the quantitative measurement available to us in the rock chip densities. The bulk densities we used are higher than the typically bulk density of unweathered granite.**

L307   There is no denudation rate data in the section titled "Morphology and denudation rate"

**removed.**

L319   Figure 1 shows the kink in the surface area profile, which I'm using as a 3 m depth scale, as occurring within the colluvium at the supposed interfluve site, and within the weathered rock in the ridge site, in contrast to the data reported here.  Please reconcile.

**Figure 1 edited. But please note that the colluvial deposit is a mobilized mass movement that happened during the glacial era. Soil formation on this deposit has been happening only since LGM. The colluvial deposit is not the same as the soil.**

L321-322  Confusing to follow trends from bottom up and from top down in the same sentence. Use a consistent direction, and perhaps break this up to be more readable.

**edited.**

L350   delete "plot of"

**edited.**

L378   Table 1 could be plotted…   The abundance of plagioclase minerals from 7 m to the surface does not appear to gradually decrease. There are also low plag numbers below 7 m.

**I plotted plagioclase on Figure 1 and let Table 1 remain.**

L417    It is very difficult to see a "gradual, uniform increase" in seismic velocity with depth in Figure 7. The color ramp emphasizes the boundary between blue and greenish colors (about 300 m/s). It would take considerable effort to analyze the hues plotted to see a uniform increase. I note that the color bar itself is non-linear.

**This color ramp is all too common in seismic survey geophysics and a different version of this figure was unavailable to the authors.**

A prominent, yet unnoticed feature in Figure 7 is the high velocity material that rises to the surface at the right end point of the transect. Please discuss this feature.

**Added, "Near the surface by Well 2 we see material of intermediate velocity, which we interpret as the colluvial deposit."**

Commonly seismic velocity profiles are used to identify fresh/weathered rock boundaries.

L438    The phrase "Weathering models expect…" anthropomorphizes weathering models.

**removed.**

L438    Please identify which weathering models produce abrupt changes in rock properties. I can think of only one—Fletcher et al. (2006, EPSL), in which a specified threshold in accumulated strain is presumed to generate fractures. This feature takes a diffusive (chemical weathering) process and produces a sharp weathering front. It's beautiful, because it matches data in way that most weathering models—whether chemical or mechanical—cannot, because diffusive processes do not produce sharp boundaries or interfaces.

**The text has been modified with the notion of identifying weathering boundaries at a specific depth, which is common in weathering studies and has been requested of this manuscript by the reviewer on several instances.**

L442    "nebulous"—consider your word choice.

**noted.**

L443    What are secondary fractures? This term suggests that secondary fractures differ from primary fractures (whatever those are), and that water behaves different in these different fractures. Please explain.

**changed to "secondary fractures are more hydraulically conductive than primary porosity"**

L447   Seismic velocity profiles tend to smear out boundaries.

**Unclear what change needs to be made here.**

L448   Could you summarize these published results in one of the figures?

**Added to figure 1.**

L449-450  I do not agree that there is a generally agreed expectation for depth of the weathering front from ridge to valley. Indeed, much of the US CZO effort was to map this boundary and understand it in as many places as possible, as a step to develop our understanding of how weathering and erosion interact.

**Changed to simply  name our finding and not state it as an expectation of weathering models.**

L455   The statement that frost and tree rooting will both produce uniform weathering to a depth of 3 meters in this landscape is conjecture. More support is needed—or label this as "we suggest" or some such phrase.

**references added.**

L465   Look at Denny & Goodlett (1956, USGS PP288, p 59) for a discussion of tree throw in this area that might be more appropriate than Roering.

**Reference added.**

L471   This paragraph does not contain any direct observations from the study site.  Rephrase this sentence—something along the lines of "These observations from other areas lead us to conjecture that the 3 m depth change in specific surface area observed in Spring Brook might possibly be related to tree rooting processes.

**edited**.

L472   Last glacial maximum is a time, not a place.

**edited.**

L475   Why are LGM periglacial processes impactful only in the continental US?  (legacies, not

legacy)

**edited.**

L479    The frost cracking window is about -3 to -8°C at whatever depth those temperature conditions occur. Hallet et al. (1991, Permafrost Perigl Process) defined the frost cracking window in lab experiments.

**reference added.**

L481    Anderson et al. (2013) are not the only authors who understand that frost cracking and frost creep are climate driven processes.  Perhaps you could cite a periglacial textbook—e.g., Washburn 1973, Periglacial Processes and Environments?

**removed.**

L483    Again, this is conjecture on the depth of frost weathering expected in this setting. Please rephrase.

**rephrased.**

L485    I don't quite see the trends described. There are no SSA-Si data from below 3 m at the interfluve site, hence no trend can be discerned.  The data from the ridge does not have a clear trend below 3 m.   Also, the wording is a little unclear given that the apparent trend is given instead as a range of values (4-10 m2/g, rather than 4 m2/g increasing to 10 m2/g from 3 m depth to the surface).

**clarified**.

L486    "supported by"?  Do you mean coincides with?   In any case, I cannot see the quantitative change in illite and vermiculate abundance referred to from the XRD patterns.  Please provide quantitative data.

**"supported by": we list the types of data that support the finding stated.**

**We have noted the illite and vermiculte increased peak intensities in the bulk XRD, which is quantitative.**

**We do not claim for our clay XRD to be quantitative. Due to the number of rinses to clean and isolate clays for XRD, we are skeptical of any reports that this method can be quantitative. This is explained in Section 4.7.**

L524-525  How does oxidation of iron-bearing primary minerals form phyllosilicates?  Please

write out some example reactions.

**Equation 2 is already provided as an example.**

L566   I think the number 2 in this line is meant to be Equation 2?

**Manuscript says Eq. 2.**

L568   See nice review of oxygen limitations in chapter by White and Buss in the Treatise on Geochemistry (2nd Ed)

**reference added.**

**Response to Reviewer #2.**

**Note: The reviewer comments are in light text, the authors replies are in bold.**

To the Editor

Dear Ed.,

I read this article with interest, as it present a different approach to study weathering profiles compare to classical studies based on geochemical or isotopic analyses. Indeed, mineral specific surface area (SSA) are used as a tool to study weathering profiles. Measurement were completed by XRD analyses in samples, induced and remanent magnetizations and seismic multichannel analysis of surface waves. This work is part of on more extensive one as the study site with two weathering profiles are part of the Critical Zone Observatory in the Pennsylvania Piedmont. The first author has previously present geochemical data from those two wells in Fisher et al., 2017a.

My first comment is linked to the introduction: Figure 1: it is not very clear for me why this figure is introduced here whereas there are elements on studied sites in it, which are introduced only after. This figure and its description should be moved at the beginning of part 3

"study site".

**In reassessing the location of this figure, which began as a conceptual diagram and migrated to a schematic data plot in revisions, we moved the hillslope figure to the results section.**

The Hypothesis (part 2) should be developed after and then only should come the Study site (part 3). In short, I would suggest reorganizing the beginning of the article and maybe re-named this part as following: Geological Setting (or another subtitle): 1) figure1 + description, 2) Hypothesis and 3) Study site.

**We think we rearranged sections according to this suggestion. In considering the suggestion we also realized that Geological Setting was a better heading for our comprehensive description of the study site and it's geological context.**

My second point is linked to the chosen method. We clearly understand the approach of the first author, who use different methods to better constrain/understand chemical weathering processes record in depth weathering profile. After using classical geochemical approached in Fisher et al., 2017b, which involve models, authors get more in details by identifying threshold within the profile. So my question is, why co-authors of this article don't use preview work done on same sites to correlate their SSA measurement to a chemical weathering index, which would corroborate transitions identify by this method in the weathering profile?

**If by Chemical Weathering index, you mean the "tau" of Brimhall & Dietrich 1987, we realized from your comment that we did not reveal how this model is connected to our existing elemental depletion measurements, so have now referenced this connection in the hypothesis.**

In addition, a minor comment: is it useful to define what is a "bedrock" (l42)? Or a "crital zone" line 36? These general definitions are given in details, which contrast with methods and results of the article, which supposed to have a specific background in chemical

weathering studies.

**We added the following to the definition: We did not encounter saprolite or saprock in our weathering profiles, which are more friable than the weathered rock we observed, so we chose to describe our rock with the more general "weathered rock" designation.**

Finally, I would like to thank you for giving me the opportunity to review this article, I would be pleased to review others in the future and as I mentioned it previously my name can appear as a reviewer for this article.

Best regards,

Sétareh Rad

Reference

Fisher, B. A., Rendahl, A. K., Aufdenkampe, A. K. and Yoo, K.: Quantifying weathering on variable rocks, an extension of geochemical mass balance: Critical zone and landscape evolution, Earth Surf. Process. Landforms, 42(14), 2457–2468, doi:10.1002/esp.4212, 2017b.

---

## Author Response (AR2)

**Comments to the author**:
Editorial Comments on Mineral surface area in deep weathering profiles reveals the interrelationship of iron oxidation and silicate weathering by Fisher et al.

Thank you for the submission of you revised manuscript. I have now read it, along with the reviews and your detailed response to the reviews. Your dataset is clearly interesting and I think you have an important finding that the SSA (effectively a proxy for secondary Fe oxide minerals) shows a continuous increase towards the surface, similar to an elemental profile, but this is decoupled vertically from non-redox sensitive mobile elemental depletion profiles. The inference being that acid base and redox reactions are happening at different depths. However, I have to agree with many of reviewer 1's comments overall and even in it's revised format this is still a challenging read and would really benefit from being substantially improve before it can be considered for publication. Putting in some more effort now will help this work achieve its full potential and get maximum citations in the future.

1. The presentation of the data really would benefit from having a single consolidated figure that illustrates the key patterns. It seems to me that one of the key observations (perhaps the most important one) is that the very nice depth trend with SSA does not mirror depth trends in either mineralogy or elemental depletion. Or rather, if I have understood, the depth trend in chemical depletion is in a different place to SSA increase. Although the chemical data is not new, I think it is really very important to see the SSA plotted alongside the elemental depletion plots from previous work. This certainly echos some of the comments from R1. Demonstrating this point clearly with a single figure seems essential before going on to consider what might be causing the differences. This is shown schematically on Fig. 2 but I would consider an overhaul of Fig 3, adding literature data to be important. The mineralogy data should presumably mirror the elemental data and that could also be plotted as a function of depth. We are told something about plagioclase showing a subtle trend on line 685, but it is never plotted.

Response: I have removed the schematic figure and and generated some new plots that show the elemental distributions with the mineralogy plots. I have plotted the SSA trends in different ways that I believe more clearly show the relationship between different data types. In devising the new plots I didn't find a logical way to put so many element distributions in the the same plot as SSA, but I think the display of data is greatly improved from this set of comments.

I struggled to understand how the change in slope in SSA was below the soil to rock boundary but above the chemical depletion boundary (line 602). This seems a little misleading if you are conceptualising this in terms of the definitions in the paragraph beginning on line 69, where the colluvium may be been transported by gravitational processes. In this case, there is a soil (or source material) but with discontinuities present. The SSA changes appear continuous across this soil to rock boundary.

Response: I'm not sure what was misleading in the original iteration, but I think the new figures, which separate data from Well 1 and Well 2 helps with clarity for the reader. I also think the new figure with drill core photos alongside SSA data for the top 4.5 m also helps the reader see where the SSA changes occur, and on this figure I have placed the soil C horizon. I hope it no longer feels misleading that the SSA changes do not coincide with the field identification of soil morphology.

2. Potential explanations are considered for the change in SSA at ~3m. One of the issues that worries me a bit about this data set and interpretation is that the SSA is made up of both Fe oxides minerals as well as phyllosillicates. Clearly oxidation should enable Fe-oxides to form from primary minerals. Mechanistically however, are no mobile elements released during the oxidation process?

Response: By adding the figures with element mass balance data for the full depth profiles, we show how Fe and Al distribute with weathering, in which both seem to undergo a small amount depletion (the depletion profiles are all scaled the same, so this turns into a relative relationship compared to the largest depletions, Ca and Na). A second figure in the discussion section shows how the iron concentration hovers very close to 10% while the Al concentration decreases across the top 4.5 meters. These data reveal that iron appears to largely be retained in the weathering profile, but aluminum seems to be partly released.

It is interesting that oxidation reactions have been conceptualised as being entirely distinct from one another. I wonder the extent to which that is correct. For example, why is there no dissolved O2 in the water that is transporting the carbonic acid to the elemental depletion front? Presumably, if you were to plot Tau_Fe, it would remain constant thoughout the profile, but the Fe is being redistributed between oxides and "primary" phases?

Response: I'm not sure which oxidation reactions were presented as distinct from one another. Regardless, dissolved O2 making it to the elemental depletion front is not something I observe in weathering or soils literature. Although many authors try to separate the biotic and abiotic components of weathering, I struggle to imagine a context in nature which O2 is not an energy source for organisms. Studies of soil and weathering ubiquitously measure and model O2 decreasing with depth. Even if O2 persists in subsurface fluid, as it penetrates deeper into the earth the concentration of O2 in the water decreases and the biological and mineralogical reactions that consume O2 are faster than diffusive process can recharge O2. (e.g. Brantley et al 2013).

3. One potentially interesting thought could relate to the kinetics of

dissolution/crystallisation. Some recent work has suggested that Fe oxides can form/recrystallise very rapidly. I wonder how this might play into the story.

It's true that oxide minerals have fast kinetics in the environment (as leveraged in stormwater and other remediation efforts), but I'm not sure how this contributes to the manuscript's main ideas of CO2/O2 and SSA. I also note that the Piedmont weathering profile has had 12,000 years to develop since the last glacial maximum, which suggests that oxide mineral kinetics are not a limitation in this system. I am concerned that adding a discussion on kinetics would be far more speculative than the discussion of oxygen as the limiting agent for weathering in this setting.

4. Are there no other literature soil profiles that have looked at Fe oxide chemical extractions? SSA data might not be available, but one might imagine similar trends might have been detected by alternative methods. This is discussed relatively briefly in the paragraph beginning on line 764, but the key thing for me is how does this compare to element depletion profiles of the mobile non-redox sensitive elements.

I added the following from two studies where I was able to find extracts of profiles:
"Other studies of soil profiles with extractable oxides removed reveal that the distribution of extractable oxides depends on soil type and morphology. Aburto et. al. 2017 studied glacial deposits in the Tahoe region in which the amount of extractable oxides increased with decreasing depth, but unlike our profiles, the extracted oxides decreased in the uppermost 30 cm. In transects in loess deposits in southern Illinois, Wilson et. al. 2013 saw extractable oxides in most profiles reach their maximum in the zone of clay accumulation (Bt horizon), with little to no decrease in extractable oxide at the 2 m depth extent of their study."

5. Tree rooting and frost damage are interesting to consider as explanations for the ~3m depth. However, fundamentally the elemental depletion profile is much deeper, suggest water, acids and likely O2 should also have been able to penetrate deeper.

The discussion of tree rooting depth has been removed from the manuscript. The point of that discussion was to address how we might facilitate an oxygen penetration deep enough to explain the extractable oxides to 3 m deep. The notion that oxygen would be likely to be deeper is not one that I see in the literature.

In summary, this is an interesting manuscript and dataset that is in the road to improvement to make a clear coherent and important study. Some additional work is required before publication.

---

## Author Response (AR3)

Associate Editor decision: Publish subject to technical corrections
by Edward Tipper
Comments to the author:
The authors have worked hard to address all the comments, and the result is much improved manuscript. This will make a good contribution and I recommend it for publication subject to some minor technical comments/questions from my side that the authors should consider:

1) Line 180: Are you sure you want to phrase this in this way? Might suggest: Although the citrate-dithionate extraction was performed on all SSA samples, the elemental composition of extracts was only determined for some intervals from Well 1.

I have applied your suggestion.

2) Line 338: This is one of several examples. Please try and adjust all examples. Rather than using phrases such as Fig X shows, or in Fig X we can see, try a phraseology similar to the following: "The concentration of total Fe (as Fe2O3) is approximately 10% along the top 4 m while the total Al (Al2O3) decreases by about 5%, to approximately 20% (Fig. 3)"
A find with the search term "figure" should help with this.

I have removed every instance of this and did the same for equations and tables.

3) Line 357: Please clarify how the iteratively segmented regression was done

Now reads, "We iteratively applied segmented regression to optimize fit and break points for each SAI profile, using SAI as the dependent variable in the regression analysis and depth as the independent variable (Figure 5). Our iterations tested the number of break points that yielded the highest $R^2$. "

4) Line 452L. Delete the space after "front" before the full stop.

Completed.

5) Do you comment on the apparent mobility of Al on Fig 6, relative to Zr? Apologies if I have missed this.

Now reads, "The elemental depletion of Al follows a similar trend as Mg and Fe, with depletion above 1.33 m in Well 1 and above 4.26 m in Well 2."